# Barcoded HIV-1 reveals viral persistence driven by clonal proliferation and distinct epigenetic patterns

Tian-hao Zhang [1,2,10,11], Yuan Shi[1,11], Natalia L. Komarova[3], Dominik Wodarz[4], Matthew Kostelny[5], Alexander Gonzales[6], Izra Abbaali[6], Hongying Chen[5], Gabrielle Bresson-Tan[6], Melanie Dimapasoc [5], William Harvey[6], Christopher Oh[6], Camille Carmona[5], Christopher Seet [7], Yushen Du[8], Ren Sun[1,9], Jerome A. Zack [5,7] & Jocelyn T. Kim [6] ✉

The HIV reservoir consists of infected cells in which the HIV-1 genome persists as provirus despite effective antiretroviral therapy (ART). Studies exploring HIV cure therapies often measure intact proviral DNA levels, time to rebound after ART interruption, or ex vivo stimulation assays of latently infected cells. This study utilizes barcoded HIV to analyze the reservoir in humanized mice. Using bulk PCR and deep sequencing methodologies, we retrieve 890 viral RNA barcodes and 504 proviral barcodes linked to 15,305 integration sites at the single RNA or DNA molecule in vivo. We track viral genetic diversity throughout early infection, ART, and rebound. The proviral reservoir retains genetic diversity despite cellular clonal proliferation and viral seeding by rebounding virus. Non-proliferated cell clones are likely the result of elimination of proviruses associated with transcriptional activation and viremia. Elimination of proviruses associated with viremia is less prominent among proliferated cell clones. Proliferated, but not massively expanded, cell clones contribute to proviral expansion and viremia, suggesting they fuel viral persistence. This approach enables comprehensive assessment of viral levels, lineages, integration sites, clonal proliferation and proviral epigenetic patterns in vivo. These findings highlight complex reservoir dynamics and the role of proliferated cell clones in viral persistence.

Despite effective antiretroviral therapy (ART), the viral reservoir persists, primarily due to the integration of HIV as provirus into host genomic DNA, forming a reservoir of infected cells. Interruption of ART can lead to the reactivation of these latently infected cells, resulting in viral rebound[1–3]. A deeper understanding of the dynamic changes underlying reservoir expansion is critical to developing a functional cure. Proviral expansion, characterized by an increase in the number of cells harboring integrated HIV DNA, is a major concern for

[1]Department of Molecular and Medical Pharmacology, University of California, Los Angeles, CA, USA. [2]Molecular Biology Institute, University of California Los Angeles, Los Angeles, CA, USA. [3]Department of Mathematics, University of California San Diego, La Jolla, CA, USA. [4]Department of Ecology, Behavior and Evolution, University of California San Diego, La Jolla, CA, USA. [5]Department of Microbiology, Immunology, and Molecular Genetics, University of California Los Angeles, Los Angeles, California, USA. [6]Department of Medicine, Division of Infectious Diseases, University of California Los Angeles, Los Angeles, California 90095, USA. [7]Department of Medicine, Division of Hematology and Oncology, University of California Los Angeles, Los Angeles, California, USA. [8]Cancer Institute (Key Laboratory of Cancer Prevention and Intervention, China National Ministry of Education), The Second Affiliated Hospital, Zhejiang University School of Medicine, Hangzhou, Zhejiang, China. [9]Center for Infectious Disease Research, School of Medicine, Westlake University, Hangzhou, Zhejiang, China. [10]Present address: Department of Molecular and Medical Pharmacology, University of California, Los Angeles, CA, USA. [11]These authors contributed equally: Tian-hao Zhang, Yuan Shi. ✉e-mail: jocelynkim@mednet.ucla.edu

therapies aimed at reducing the latent reservoir burden. This expansion can occur through two main mechanisms: increase of the number of proviruses by de novo infection of different target cell clones and proliferation of cell clones already containing proviruses. De novo infection of target cells is certainly important during primary infection to seeding proviral DNA and the viral reservoir[4,5]. During ART suppression when viral infection is halted, it is evident that infected cell clones can proliferate and expand the proviral reservoir[6,7]. Proliferation of cell clones harboring provirus can also occur during early infection in HIV-1 infected humanized mice and early infection with persistence after ART initiation in at least two individuals with HIV[8,9]. Further research is needed to better understand the relative contributions of viral seeding and cell clonal proliferation to proviral expansion during different phases of HIV infection and treatment. Developing high-throughput analytical techniques capable of simultaneously and accurately measuring both mechanisms will be crucial for dissecting the complexities of reservoir dynamics and informing the development of targeted therapeutic strategies.

Persistent proliferated cell clones harboring proviruses represent a significant area of study in HIV research. Understanding whether these proliferated cell clones are prone to elimination by selective forces is crucial for deciphering the dynamics of the viral reservoir. Despite effective ART, proviral transcription among infected cells persists[10,11] and can induce cytotoxicity or anti-viral host immune responses. Recent studies have shown an accumulation of intact proviruses in heterochromatin among elite controllers and people living with HIV/AIDS (PLWH) who are on over two decades of long-term suppressive ART[12,13]. In addition, a recent study demonstrated CD4 + T cells harboring intact proviruses had significantly decreased inherent cellular proliferation capacity upon ex vivo TCR stimulation compared to uninfected or infected cells harboring defective proviruses[14]. Indeed, the relative proportion of intact proviruses has been shown to decline in comparison to defective proviruses[15]. However, despite mechanisms that drive reduction of the viral reservoir, it has been estimated to take 226 years of effective ART to eliminate intact proviral DNA by 4 logs[16]. There are multiple genomic and epigenetic factors that can contribute to the persistency of latent intact proviruses, but it is hard to generalize a consistent rule in predicting which infected cells harboring intact proviruses are most likely to persist. For one thing, in human HIV infection, a significant bottleneck effect typically results in a single or few viral variants, termed founder viruses, successfully establishing and expanding within the host. This complicates analysis of viral lineage changes and their seeding capabilities of proviral DNA, as different lineages may emerge at varied times experiencing distinct evolutionary trajectories. In addition, analysis of proviral integration sites suggests a selection for repressive chromatin features. However, it is hard to analyze these epigenetic features in a systematic way, likely caused by the sparsity and clonality of proviral integration sites in these samples. So far, there is no robust and simple model and platform to analyze viral lineage at rebound and its association with the respective proviral integration site features among proliferated and non-proliferated cell clones.

Understanding the fate of proliferated cell clones and their contribution to viral persistence is essential for developing targeted therapeutic strategies. Recent evidence suggests that integration of intact proviruses into oncogenes is not frequently observed in proliferated cell clones[17]. However, transcriptionally active proviruses in genes associated with cell survival and proliferation have been detected[13,18,19]. Antigen-mediated proliferation has been characterized in CD4+ T cell clones from ART-suppressed people[20] and an individual with HIV and squamous cell cancer, who had a persistent cell clone contributing to low-level viremia and likely driven to proliferation by tumor antigen levels[21]. General immune activation[22,23] and homeostatic proliferation[24,25] may also contribute to

proliferation of infected cell clones. Regardless of the mechanism of proliferation of infected cell clones, others have found the proviral sequences before analytic treatment interruption (ATI) matched rebounding plasma sequences after ATI[26,27] with clonal proviral populations suggesting high viral seeding or cell clonal proliferation. Proliferated cell clones prior to ATI were the origin of rebound viremia in at least one individual using simultaneously sequencing of the proviral genome and matched integration site as well as TCR sequencing from ex vivo stimulated CD4 + T cells[19]. These studies show that persistent proliferated cell clones can potentially fuel rebound viremia after ATI. However, the generalizability of these cases is not clear. Proliferated cells may still undergo active proviral transcription and be eliminated by virus-related cytotoxicity or host immune system selection. The accumulation of proviruses with repressive chromatin features could lead to a reservoir in a temporary "locked" state, such that no transcription or antigen driven elimination can purge those lineages during ART. For those cells, how temporary the "locked" states are, and under what circumstances this repressive epigenetic status will be reverted are important question to answer in achieving post treatment control of disease. A more detailed epigenetic picture of proviral integration sites with known post ART fates are essential to answer these questions.

Various methods have been employed to quantify proliferated cell clones harboring provirus by shearing of genomic DNA to generate rare DNA fragments harboring integration sites of varying breakpoint lengths[6,28] or limited dilution assays for single integration site amplification[7,29,30] with parallel detection of viral RNA transcripts[18]. While these approaches have been used to detect cellular clonal proliferation, there are limitations in sensitivity and resolution in distinguishing multiple proviral integration events within a proliferated cell clone accurately. Thus, it has been difficult to precisely measure the size of proliferated cell clones and accurately compare populations of proviral lineages among proliferated versus non-proliferated cell clones in the reservoir. It is therefore important to develop an assay that can simultaneously follow proviral lineages and integration sites, while efficiently tracking viral RNA lineages and cell clonal proliferation in parallel has not been described.

In this work we address important questions regarding the dynamics of the HIV reservoir and the mechanisms underlying viral persistence, with potential implications for the development of strategies to achieve a functional cure for HIV infection. In order to accurately detect proviral integration sites and associate these proviral lineages with post-ART rebound capabilities, we introduced a synthetic genetic barcode into the genome of a full-length R5-tropic HIV-1 NFNSX, which replicated efficiently and formed a reservoir in humanized mice[31]. By employing PCR and deep sequencing methodologies, we developed a highly sensitive and high-throughput approach to quantify single viral RNA and DNA molecules to the resolution of a single copy in parallel, allowing for the multidimensional assessment of viral seeding, cell clonal proliferation, proviral expansion, and virus production. We show the proviral reservoir maintains genetic diversity despite perturbations such as cellular clonal proliferation and viral seeding by rebounding viral lineages. We find proliferated, but not massively expanded, cell clones contributed to proviral expansion and was strongly associated with viremia and viral seeding. Among non-proliferated cell clones there was an apparent elimination of proviruses associated with transcriptional activation and viremia. However, proliferated cell clones harboring proviruses were persistently associated with activating epigenetic marks, suggesting that proviruses in proliferated cell clones could drive viral persistence and viremia. The current work provides a detailed analysis of viral levels, lineages, integration sites, clonal proliferation and proviral epigenetic patterns during reservoir dynamics in vivo.

## Results

### Construction of a replication-competent barcoded CCR5 (R5)-tropic HIV-1

To track viral lineage changes over time from the beginning of infection establishment in vivo, each recombinant clone was labeled with a unique molecular sequence, so that its population dynamics could be retrieved by sequencing. Here we refer to each viral lineage as a viral barcode. We utilized molecular barcoded technology to create a barcoded isogenic swarm of R5-tropic HIV NFNSX (NFNSX-BC)[31] tagged with distinct 21 bp barcodes as a way to track viral lineages via their molecular barcodes (Figs. 1a and S1a-d). The resulting recombinant NFNSX-BC contained similar levels of HIV p24 compared to the parental non-barcoded NFNSX as quantified by ELISA (Fig. S1e), infected the reporter cell line GHOST CXCR4+CCR5+ cells comparably (Fig. S1f). We also found viral production was similar from GHOST (3) CXCR4 + CCR5+ cells infected with the barcoded and non-barcoded NFNSX in vitro (Fig. S1g). Additionally, we injected equal doses of NFNSX-BC and non-barcoded NFNSX into humanized mice and found comparable viral loads during primary infection (Fig. S1h), indicating similar in vivo replication kinetics of the barcoded and non-barcoded HIV strains. These results indicate the barcode sequences likely posed no selection pressure to the virus replication in vitro or in vivo.

Next, to assess the in vivo replication of NFNSX-BC, we intravenously injected 500 ng p24 of NFNSX-BC virus into humanized BLT TKO mice[32]. We selected three timepoints to perform endpoint analysis: early infection (six weeks after HIV injection), ART suppression (six weeks after ART treatment), and rebound infection (six weeks after ART interruption) (Fig. 1b). ART was comprised of raltegravir, emtricitabine, and tenofovir disoproxil fumarate, which was administered in the animal feed[33]. Plasma HIV RNA copies were measured by qRT-PCR (Fig. 1c) as well as frequencies of peripheral human CD45 and human CD4 + T cells by flow cytometry (Fig. S2a). There was no significant differences in the frequencies of human CD45+ cells in the blood between groups of mice (Fig. S2a, *left*). As expected, the frequencies of CD4 + T cells significantly declined after 6 weeks of HIV infection, was reversed by ART, but then declined again during rebound infection (Fig. S2a, *right*). At endpoint analysis, total HIV RNA and DNA levels were measured by qPCR from the organs of spleen, bone marrow, and human thymic implant (Fig. S2b, c). Similar to what has been observed previously with wild type NFNSX, we found NFNSX-BC rebounded efficiently to a copy number of approximately $10^4$ copies per ml in peripheral blood, spleen, bone marrow and human thymic implant[33]. Cell-associated HIV RNA and total HIV DNA levels also followed similar dynamics as expected, in which viral loads were significantly higher in mice during early and rebound infections compared to ART suppression, due to active viral replication and viral seeding of proviral NDA.

### Proviral barcode is linked to integration site simultaneously via PCR

We next tracked each viral lineage by barcode sequencing, and further identified its integration site in its proviral DNA form. Limiting dilution assays[29] or single cell sequencing of rare sorted p24+ cells[19] can be used to detect single molecule integration sites and their matching proviral sequences. However, translating methods based on limiting dilution assays to humanized mouse samples can be challenging due to the limited number of cells available. This often necessitates high-throughput, labor-intensive methods to avoid undersampling of the reservoir. To link viral barcode information with its integration sites and quantify each barcode precisely, we developed a nested PCR and linkage method to deep sequence HIV DNA barcodes, and, if present, their matching integration sites (BI-seq). Using a unique molecular identifier (UMI) to label each DNA molecule during the library preparation, we tracked both proviral lineage by their viral barcode and cellular lineage by their integration site at the single molecule level. Specifically, we extracted tissue genomic DNA and subjected the DNA

to enzymatic fragmentation, end repair and adapter ligation, so that each DNA molecule was labeled with UMI (Fig. S3a, b). In addition, regions that contained the proviral barcode and integration site were linked and amplified by PCR (Fig. S3b). The analysis workflow and typical proviral DNA barcode quantification and linkage was provided (Fig. S3a–c). Compared to standard qPCR, we found the UMI count of HIV DNA obtained using BI-seq significantly correlated within the viral load range of 4 logs (Fig. S4a) and was more sensitive, detecting as few as 2 viral DNA molecules (Fig. S4b) with high reproducibility (Fig. S4c–e).

### Stable diversity of proviral barcodes over time

From the tissues of 20 NFNSX-BC infected animals, we performed viral barcode analysis on extracted RNA and genomic DNA in parallel. For the viral RNA barcode analysis, we used our previously described methods on extracted RNA of the plasma and tissues[33,34]. Due to undetectable or minimal HIV RNA loads during ART suppression, the RNA barcodes were only prepared from early and rebound infections. There was a severe genetic bottleneck after intravenous injection of NFNSX-BC, restricting the number of viral RNA barcodes that established early infection in vivo with a subsequent loss in number and diversity and increase in dominance of rebounding viral RNA barcodes after ART was discontinued (Fig. 1d–g, Supplementary Data S1), which was similar to previous in vivo studies using a barcoded CXCR4 (X4)-tropic and vpr-deficient HIV-1[33,34]. Using BI-seq as a parallel platform, we examined a total of 292,576 viral DNA molecules from extracted genomic DNA of which 7.35% was integrated (Fig. S5a, b). We found the total number, diversity, and dominance of proviral DNA barcodes were similar between early infection, ART suppression, and rebound infection (Fig. 1h–k). Recent work shows that the genomic diversity of proviral lineages is stable on long-term ART[35]. We found here that even during early infection and ART suppression, despite dominant viral barcodes that emerged and seeded the reservoir, the genetic diversity and stability of proviruses were still maintained.

### Viral seeding spreads HIV during early and rebound infections

Persistent HIV infection leads to de novo infection or clonal expansion of cells harboring proviral lineages. Both events can be quantified by our platform since we can track both viral lineages and cell clones simultaneously with high sensitivity. Seeding events can lead to the establishment of an independent integration site, and thus the number of integration sites per barcode (IS per BC) represents the seeding efficiency of each viral barcode into proviral DNA (Fig. 2a), which we refer to as the viral seeding score in the following writing. Integration sites were identified by aligning the sequences to the human genome (hg38, Ensemble release 108) from the USCSC Genome Browser database[36]. We detected 15,305 integration sites, which were linked to 504 proviral barcodes, and found some proviral barcodes seeded into thousands of distinct cell clones (Fig. 2b, Supplementary Data S2), highlighting the sensitivity of our approach to quantify the size of proviral barcodes. Because a viral lineage can infect many different cell clones in different tissues during early infection of PLWH[4], we hypothesized that the viral seeding score would be higher during early and rebound infections compared to ART suppression. As expected, the viral seeding score was significantly higher during early and rebound infections compared to ART suppression, indicating that viral barcodes infected many cell clones during periods of active viral replication. The seeding score of a viral barcode that was detected during ART suppression likely reflected the viral seeding that occurred prior to ART initiation with losses of viral barcodes during ART. Rare proviral barcodes were likely eliminated during ART and could not be replenished despite vigorous viral seeding that was re-initiated during rebound infection. The viral seeding score detected during rebound infection likely

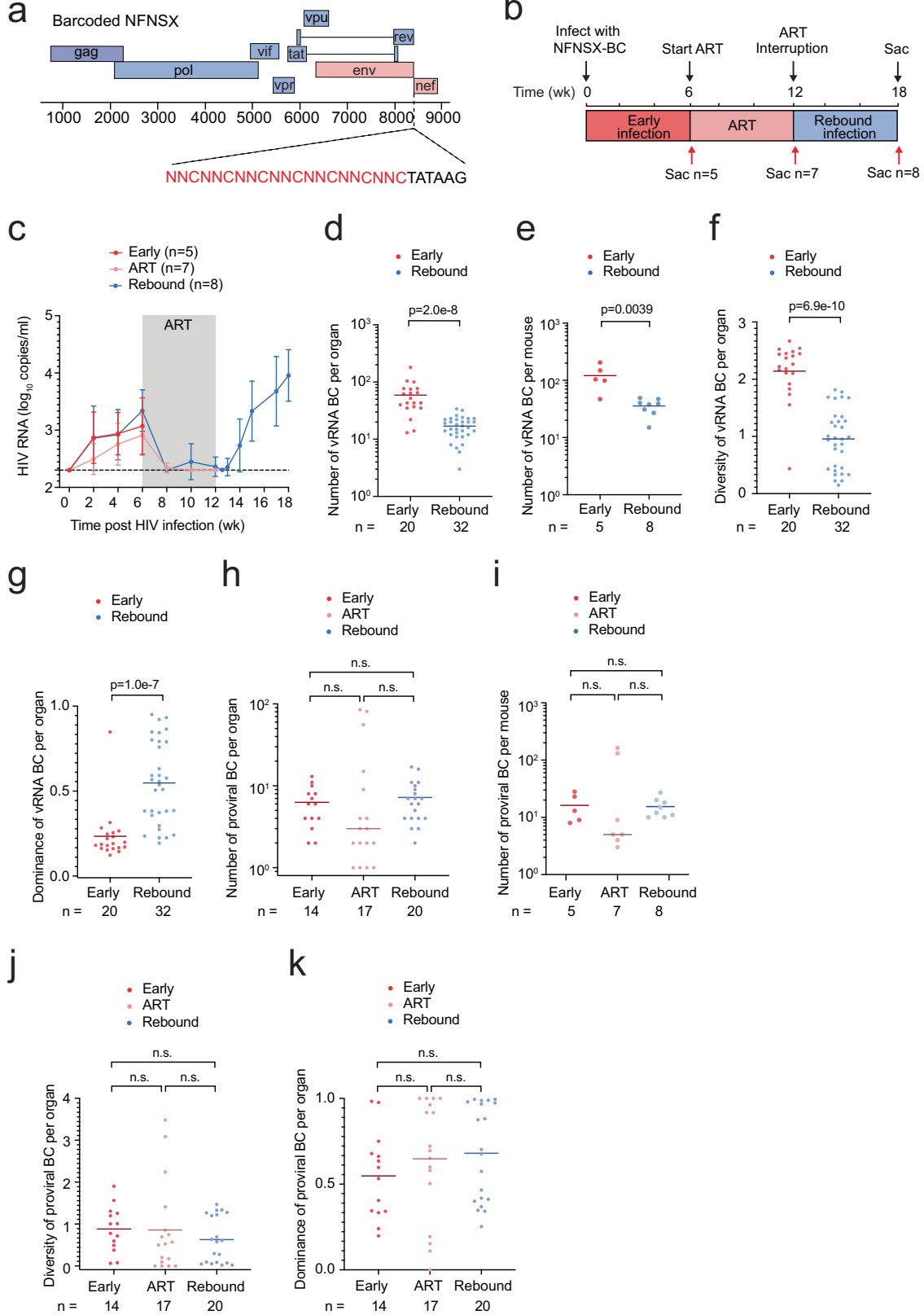

reflects the viral barcode seeding and reseeding during early and rebound infections, minus the viral barcodes lost during ART.

We next sought to understand how viral seeding could contribute to viremia generation in animals during the infection course. Since we could track viral barcodes in the viral RNA form in peripheral blood, we defined a provirus as associated with viremia if there

was a matching viral RNA barcode detected in the plasma of the same animal. The proviruses classified as associated with viremia could be contributing to active virus production and/or represent seeding of that circulating viral barcode. Proviruses classified as not associated with viremia were unlikely to be contributing to active viremia but could represent infection by an earlier viral barcode variant, which

**Fig. 1 | Barcoded full-length R5 tropic HIV-1 forms a reservoir and reveals stable genetic diversity of proviral lineages. a** Schematic showing cloning approach of the insertion of a 21 nt genetic barcode tag (red) constrained by a cytosine every third nt inserted downstream of env in a full length R5-tropic HIV-1 isolate NFNSX[31]. **b** Schematic representation of experiment; early infection of TKO-BLT mice with NFNSX-BC for 6 weeks, ART for 6 weeks, then ART interruption, followed by 6 weeks of monitoring for rebound infection. The red arrows denote when animals were sacrificed during early infection, ART suppression, and rebound infection. **c** Longitudinal plasma HIV RNA loads of groups of animals that were sacrificed after early infection, ART suppression, or rebound infection. Gray shading indicates an

ART treatment period. The black dashed line indicates the detection limit of 2.3 log RNA copies per ml. Median ± interquartile range. **d, e** Number of viral RNA (vRNA) barcodes (BC) by timepoint. *n* represents the number of organs (**d**) or mice (**e**). **f, g** Shannon's diversity (**f**) and Simpson's dominance index (**g**) of vRNA BC by timepoint. *n* represents the number of organs. **h, i** Number of proviral BC by timepoint. Horizontal bars represent medians. *n* represents the number of organs (**h**) or mice (**i**). **j, k** Shannon's diversity (**j**) and Simpson's dominance index (**k**) of proviral BC by timepoint. *n* represents the number of organs. Horizontal bars represent mean values (**d–g, j, k**). n.s., *p* > 0.05. *P* values were calculated using the two-tailed Mann-Whitney test (**d–k**). Source Data **c–k**.

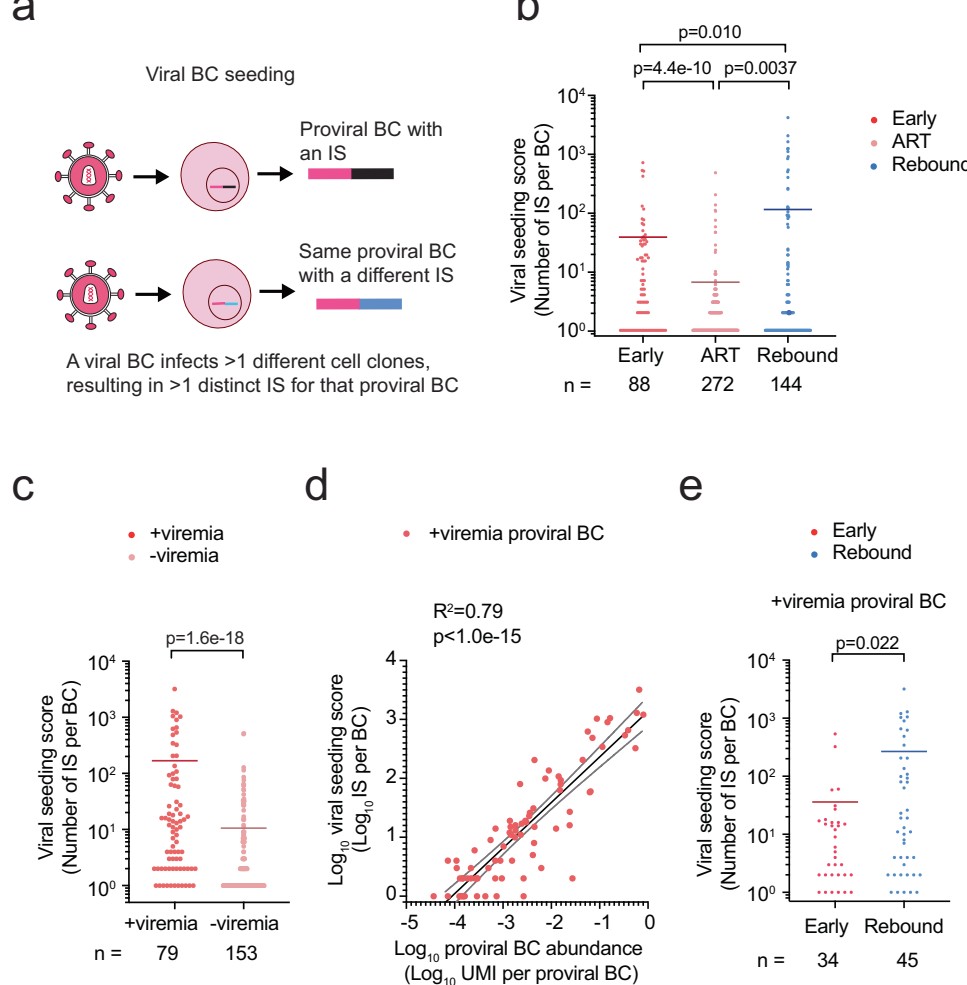

**Fig. 2 | Viremia is associated with high viral seeding of the proviral reservoir. a** Schematic depicting viral barcode (BC) seeding is measured by the integration site per barcode (IS per BC). **b** IS per BC by timepoint. *n* represents the number of proviral BC. **c** IS per BC for each provirus associated with (+viremia) or without (-viremia) viremia. *n* represents the number of proviral BC. **d** Among the +viremia proviruses, linear regression of log transformed UMI per BC and the number of IS

per BC. Coefficient of determination, $R^2$, and associated *p* value. *n* = 79. **e** Among +viremia proviruses, IS per BC during early or rebound infection. *n* represents the number of proviral BC associated with viremia. **b, c, e,** Horizontal bars represent mean values. n.s., *p* > 0.05. *P* values were calculated using the two-tailed Mann-Whitney test. Source Data **b–e**.

may have circulated, but was no longer present in the plasma at time of sample collection. We did not include proviruses during ART suppression in this classification, because we could not accurately determine which proviruses would be associated with viremia. We investigated 232 proviral barcodes during early and rebound infections and found proviruses associated with viremia demonstrated a higher viral seeding score compared to proviruses not associated with viremia (Fig. 2c). Among the proviruses associated with viremia, there was a significant and positive relationship between the viral

seeding score and the abundance of a proviral barcode (UMI per proviral BC), suggesting that viral seeding contributed to proviral expansion during periods of active viral replication (Fig. 2d). In addition, among the subset of 79 provirus barcodes associated with viremia, the viral seeding score was significantly higher during rebound infection compared to early infection (Fig. 2e). The higher viral seeding score during rebound infection was most likely due to emergence of dominant viral barcodes during rebound infection, which likely reseeded the reservoir.

## Proliferation-competent cells harboring provirus survive ART

We next assessed clonal expansion of cells harboring provirus during the different stages of HIV infection in vivo. Cellular clonal expansion of provirus-harboring cells leads to the replication of cellular clones, which was represented as integration sites in our BI-seq platform. Importantly, due to the sensitivity of our methods we could also measure the UMI per integration site (UMI per IS), which reflected the size of each provirus-harboring cell clone (Fig. 3a). We analyzed a total number of 21,296 cells harboring provirus and classified 7,075 (33.2%) as proliferated cell clones based upon a UMI per IS > 1 (Fig. 3b). Additionally, we observed a high accuracy of predicting the fraction of proliferated cell clones in our samples, which depended on the sample size for all the organs at each timepoint (Fig. S6a, Supplementary Note 1, Supplementary Data S3, Fig. S6b–9). The proportion of infected cells from proliferated versus non-proliferated cell clones was significantly highest during early infection compared to ART suppression or rebound infection (Fig. 3c). During ART suppression, we found 31% of infected cells were from proliferated cell clones, which was similar with the frequency of proliferated cells harboring provirus in ART-suppressed PLWH[6].

The proportion of proliferated infected cells was significantly lower during ART suppression compared to early infection (Fig. 3c), which can be caused by two reasons: 1) elimination of infected cells after ART initiation, and 2) a decreased cellular proliferation capacity. Because most infected cells harboring provirus are eliminated and only a subset of memory resting CD4 + T cells survive to persist during ART[24,37], we hypothesized the size of infected cell clones might be stable throughout early infection and ART suppression. We analyzed 15,305 integration sites or infected cell clones and found 1,084 (7.08%) cell clones had more than one cell (UMI per IS > 1) (Fig. 3d) with some cell clones comprised of >400 cells (Fig. 3e). These results suggest that some of the infected cell clones are large in size, thus explaining how several thousands of cells harboring provirus could be represented by a small (~7%) fraction of cell clones. Among the 1,084 proliferated cell clones, we quantified the size of the cell clone at each timepoint (Fig. 3e). The size of the cell clones was not affected by ART, suggesting the decline in the frequency of proliferated infected cells between early infection and ART suppression may not be due to declining cell proliferative capacities, but the elimination of infected target cells after ART initiation. Although the size of infected cell clones was similar during early infection and ART suppression, there was a slight but significant drop during rebound infection. The decrease in the size of the provirus-harboring cell clones during rebound infection could reflect either selective elimination of proliferated proviral cells or reduced cellular proliferative capacity[11,29,38,39].

## Cellular clonal proliferation contributes to proviral expansion and viremia

Next, we assessed the contribution of cellular clonal proliferation to proviral barcode expansion. We found a positive correlation between the frequency of proviral barcodes and the size of the provirus-harboring cell clones, indicating the frequency of barcodes were associated with proviral cell division in all three stages of HIV infection (Fig. 3f–h). Thus, the propagation of proliferated provirus-harboring cells outcompeted its complete elimination and became a driving force in determining the success of a viral lineage.

We then examined whether proviral barcodes associated with viremia correlated to cellular clonal proliferation. Because most productively infected cells die rather than proliferate, we expected that proliferated cell clones harboring proviruses associated with viremia might be difficult to detect during periods of active viral replication. However, we found proviral barcodes associated with viremia and seeding were approximately 8.8 times more likely to be in at least one cell clone that proliferated (UMI per IS > 1) compared to proviral barcodes that were not associated with viremia (Fig. 4a), indicating

proviruses associated with viremia strongly correlated with cellular clonal proliferation. We next examined this correlation during early and rebound infections. Proviral barcodes associated with viremia were approximately 4.18 and 19.7 times more likely to be in proliferated cell clones compared to non-proliferated cell clones during early and rebound infections, respectively (Fig. 4b, c). These results suggest proviruses associated with viremia were preferentially in cell clones that were likely to proliferate or that proliferated cell clones were actively contributing to viremia.

Because we used integration sites to identify cell clones, cellular proliferation must occur after integration. Thus, it is possible that proviral activation with virus production in cell clones could trigger cellular proliferation. For example, cells harboring proviruses near transcriptionally active regions could be more likely to generate higher innate immune response[40,41], which in return could serve as autonomous cues to promote T cell proliferation. However, we found proliferated cell clones harboring proviruses associated with viremia were smaller in size compared to those not associated with viremia (Fig. 4d). We also found highly proliferated cell clones were associated with lower viral seeding scores during early infection with a similar but non-significant trend during rebound infection (Fig. 4e). Thus, these results indicate proliferated cell clones were highly associated with viremia, but an increase in cell clone size was negatively correlated with viremia and viral spread. Indeed, recent ex vivo work showed that CD4 + T cells harboring intact proviruses may have lower cellular proliferated capacities compared to CD4 + T cells harboring defective proviruses[14]. It is also possible these results could indicate that proviruses near transcriptionally active regions induced cellular clonal proliferation, but some of these proliferated cells were preferentially eliminated.

## A minor proportion of proviruses associated with cell clonal proliferation were in genes associated with cancer or cell survival

We next assessed whether integration into oncogenes could contribute to cell clonal proliferation. We compared the alignments of 15,305 integration sites to the human genome and the list of genes associated with cancer based on the Catalogue of Somatic Mutations in Cancer (COSMIC) Cancer Gene Census (CGC)[42]. Only a small fraction (475 of 15,305 IS, 3.10%) of integration sites were enriched in cancer-associated genes compared to the normal frequency of genes (protein coding and characterized long non-coding RNA) in the human genome (736 cancer genes out of 25,660 total genes, 2.10%)[7] (Fig. 5a, see Supplementary Data S4). We also compared the distribution of integration sites among proliferated and non-proliferated cell clones and found a similar distribution in cancer related genes from the COSMIC reference data set (Fig. 5b, Supplementary Data S4). Next, we assessed if there were any other specific genes enriched among the integration sites from proliferated cell clones versus non-proliferated cell clones. Although we found proviruses were in previously reported genes associated with cell proliferation or survival (BACH2, MKL2, STAT5B)[6,7,43], we found eight genes (e.g. GUCY2EP, WWC1, TNFRSF10D, FOXP4, HYDIN, TNFRSF01B, NBPF12, PMS2P3, and ENSG00000261502) were significantly enriched among proliferated cell clones compared to non-proliferated cell clones (Fig. 5c, Supplementary Data S5). Five (e.g. WWC1, TNFRS10D, FOXP4, TNFRS10B, NBPF12) out of these eight genes were associated with tumor progression, cell proliferation and anti-apoptosis. However, these five genes were detected in only 3.6% of proliferated cells (776 out of 21,296 proliferated cells) and 8.4% of all the proliferated cell clones (91 out of 1,084 integration sites) (Fig. 5d), which was similar to a previous report that only 6.95% of integration sites among proliferated clones were in oncogenes in HIV-infected humanized mice[44]. Further only 2.3% (489 out of 21,296 proliferated cells) of proliferated cells had integration sites that were in the same orientation as their gene

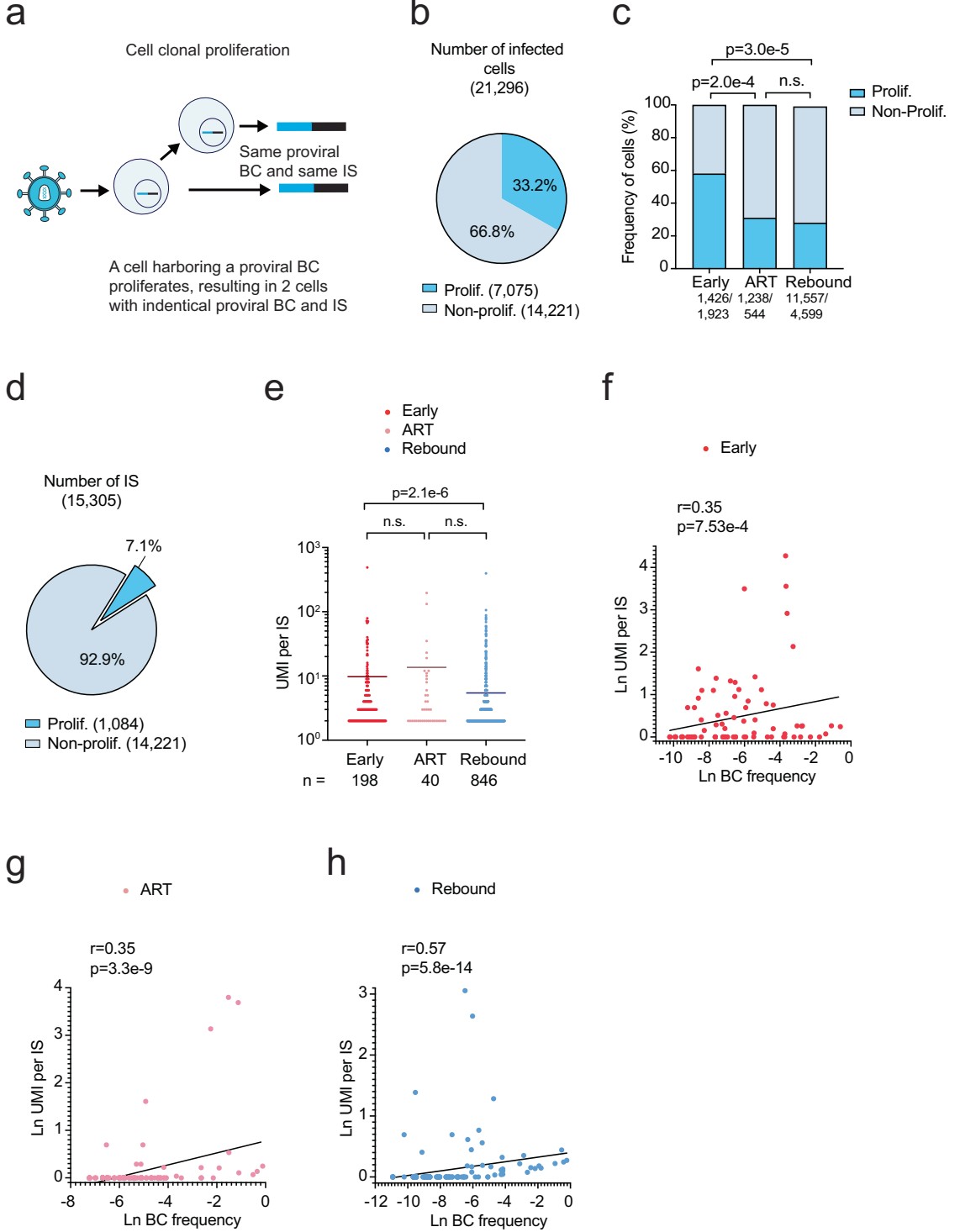

**Fig. 3 | Cellular clonal proliferation contributes to proviral expansion.**
**a** Schematic depicting proliferation of infected cell lineages measured by UMI per
IS. **b** Contribution of proliferated and non-proliferated cells among all infected cells
harboring provirus. **c** Contribution of proliferated cell clones harboring provirus by
timepoint. **d** Contribution of proliferated and non-proliferated cells among all
infected cells harboring provirus. **e** Cell clone size (UMI per IS) for each proliferated
cell clone by timepoint. Horizontal bars represent mean values. n.s., $p > 0.05$. P

value was calculated using the two-tailed Mann-Whitney test. $n$ represents the
number of IS. **f**–**h** Correlation of natural log transformed proviral barcode fre-
quency and UMI per IS per sample for each timepoint among the proviral barcodes
in proliferated cell clones during early infection (**f**), ART suppression (**g**), or
rebound infection (**h**). Black trendline line calculated by simple linear regression.
Spearman's rank correlation coefficient (r) and associated $p$ values. Early, $n = 88$;
ART, $n = 272$; Rebound, $n = 144$. Source Data **b**–**h**.

(WWC1, FOXP4 and NBPF12). These results suggest only a small frac-
tion of proliferated cell clones had integration sites enriched in cell
survival genes, indicating a modest effect of integration in cell survival
genes on proviral distribution.

Because our BI-seq approach accurately measured the size of
provirus-harboring cell clones, we assessed whether any of the pro-
liferated cell clones had integration sites in genes associated with
cellular proliferation or survival might demonstrate enhanced cellular

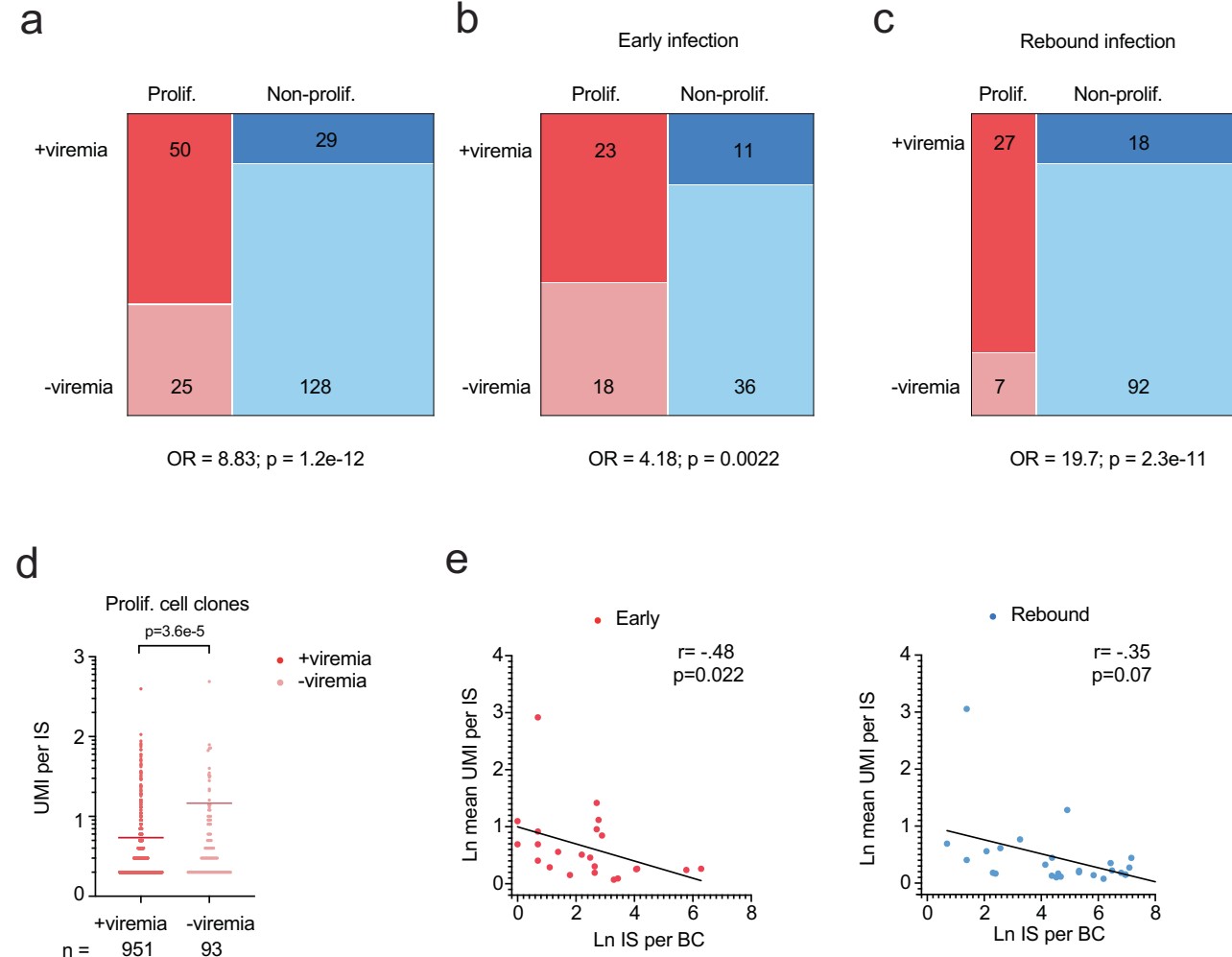

**Fig. 4 | Viremia is strongly associated with proliferated cell clones of smaller size. a–c** Mosaic plots of the number of +viremia and -viremia proviruses associated with proliferated and non-proliferated cell clones during early and rebound infections (**a**), early infection only (**b**), or rebound infection only (**c**). Odds ratio (OR) of detecting proviruses associated with cellular clonal proliferation among proviruses associated with or without viremia. *P* value was calculated by two-sided Fisher's exact test. **d** Among proliferated cell clones, the UMI per IS of cell clones harboring +viremia or -viremia proviruses. *n* represents the number of IS among proliferated cell clones. *P* value was calculated using the two-tailed Mann-Whitney test. **e** Correlation of natural log transformed mean cell clone size (UMI per IS) and viral seeding (IS per BC) among +viremia proviruses in proliferated cell clones during early (*left*) and rebound (*right*) infections. Black trendline line calculated by simple linear regression. Spearman's rank correlation coefficient (r) and associated *p* values. Early, *n* = 23; Rebound, *n* = 27. Source Data Fig. **a–e**.

proliferation. Interestingly, we found proliferated cell clones with integration sites in two genes (TNFRSF10D and FOXP4) were significantly larger in clone size (Fig. 5e). However, only 4.6% (50 out of 1,084 integration sites) of the proliferated cell clones and 2.3% of all proliferated cells (490 out of 21,296 proliferated cells) had integration sites in WWC1 and TNFRSF10D and demonstrated enhanced cell clone size. Further, only 1.2% (251 out of 21,296) of proliferated cells had integration sites that were in the same orientation as a gene associated with cellular proliferation and survival (WWC1) and demonstrated increased cell clone size. Thus, these data suggest for the majority of cell clones post-integration selection was likely not driving cellular proliferation and survival.

**Integration sites are enriched in genes and genome regulatory regions**

Because of the strong correlation between cellular clonal proliferation and viral seeding, we suspected proliferated cell clones harbored certain genetic or epigenetic features associated with transcriptional activation and virus production rather than transcriptional silence or a "locked" state. With integration site information at hand in our study,

we first performed a deeper analysis of the genomic landscape by aligning 15,305 integrate sites to the hg38 Ensemble release 108[45] (Fig. S10a, Supplementary Data S4). We found 62.4% of the integration sites were in genes, 6.9% in promoters, 13.8% in enhancers, 5.3% in CCCTC-binding factor (CTCF) sites, 0.06% in transcription factor (TF) binding sites (Fig. S10b). We were able to accurately map 15,294 integration sites to the RepBase database[46], which contains a curated and detailed annotation of different classes of repetitive DNA elements in the human genome including long interspersed nuclear element (LINE), short interspersed nuclear element SINE, long terminal repeat (LTR), satellite DNA, DNA and simple sequence repeat (SSR) regions. Approximately, 42.5% of the integration sites were in repetitive DNA elements (Fig. S10c), of which 54.2% were in non-genic regions (Fig. S10d). Compared to a randomly generated control, all the proviruses were 1.17-fold more likely to be detected in genes and were 1.37 to 2.85-fold more likely to be in genome regulation elements such as promoters, enhancers, CTCF or TF binding sites (Fig. 6a, *left column*).

We also compared epigenetic features of 15,305 integration sites by aligning them to the annotated and well-defined chromatin immunoprecipitation sequencing (CHIP-seq) deep sequencing data of

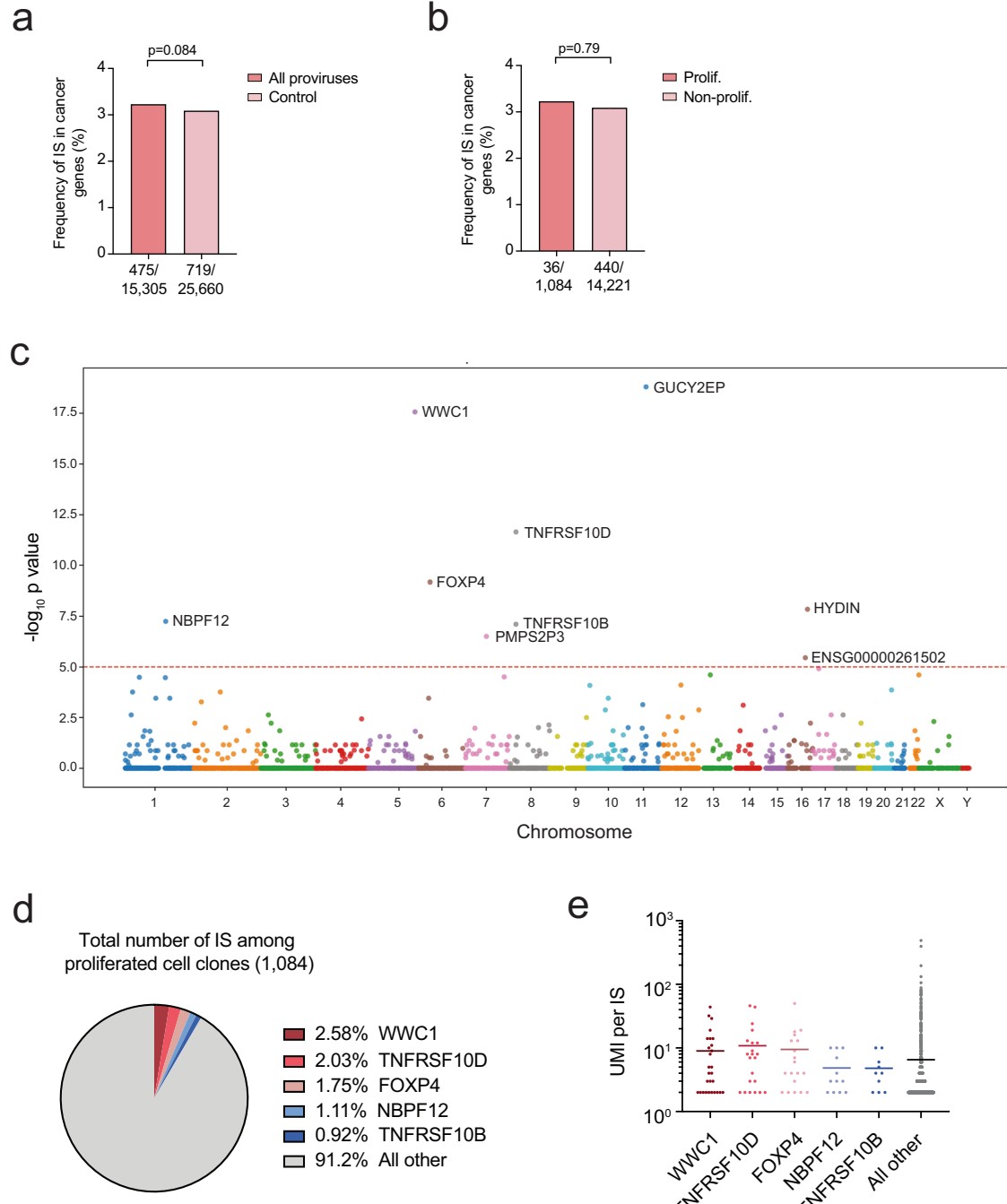

**Fig. 5 | Minor proportion of proviruses associated with cell clonal proliferation were in genes associated with tumorigenesis. a**, **b** Frequency of IS location in genes related to cancer based on COSMIC data base between all proviruses and the normal distribution of these genes in the human genome (**a**) or proviruses in proliferated versus non-proliferated cell clones (**b**). *P* value was calculated using the two-tailed Mann-Whitney test. **c** Dot plot representing the integration events that were enriched in genes among proliferated versus non-proliferated cell clones. Odds ratio and *p* values were calculated by two-sided Fisher's exact test. Y-axis indicates the *p* value (-log$_{10}$). X-axis indicates the chromosome number and position. Red dashed line indicates the threshold for statistical significance after Bonferroni correction of *p* value of 0.05. **d** Contribution of integration sites in the listed genes among proliferated cell clones. **e** Among proliferated cell clones, the cell clone size (UMI per IS) for each integration site. Horizontal bars represent the mean. TNFRSF10D vs all other, *p* = 0.0044; FOXP4 vs all other, *p* = 0.0178. WWC1, NBPF12, or TNFRSF10B vs all other, *p* > 0.05. The adjusted p values were calculated by comparing the UMI per IS for each of the listed gene to the UMI per IS for all other genes by Kruskal-Wallis test. Source Data **a**–**e**.

resting primary CD4 T cells histone modifications in the Encyclopedia of DNA element (ENCODE) regions[47]. Consistent with previous findings, we found all integration sites were 1.3 to 4.1 more likely to align inside histone modification regions associated with viral transcription such as H3K4me3 and H3K27ac, H3K4me1, and H3K36me3 (Fig. 6a, *left column*)[48,49]. These results were consistent with previous studies demonstrating HIV favors integration in genes and active transcriptional units[48,50,51]. As expected, the alignment did not reveal significant enrichment of the inhibitory histone modification H3K9me3, which is associated with constitutive heterochromatin and transcriptional suppression. However, integration sites were 3.2-fold more likely to align to areas enriched for the histone modification H3K27me3, which

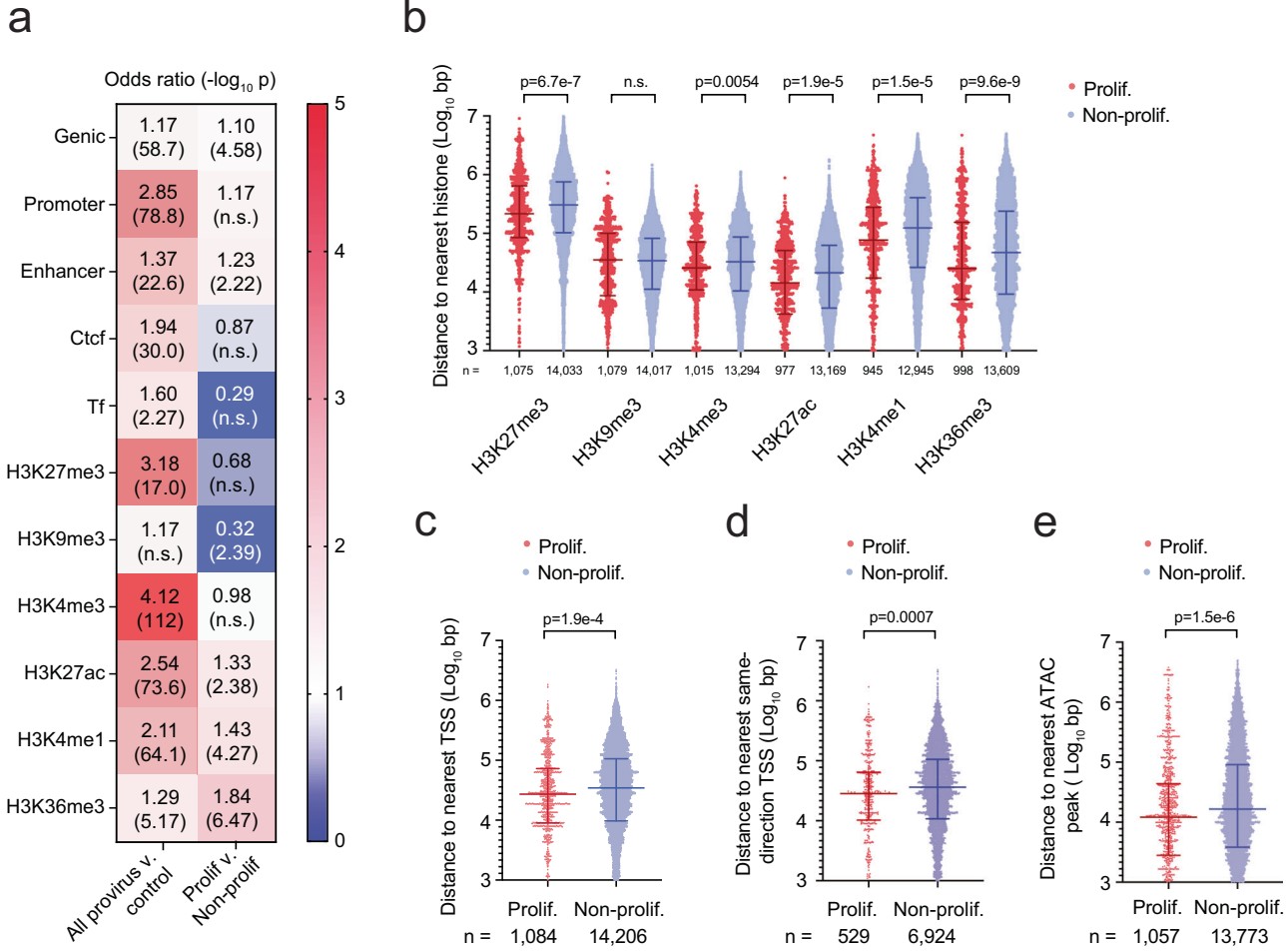

**Fig. 6 | Proviruses in proliferated cell clones are associated with activating chromatin features. a** Heat map demonstrating the odds ratio of proviruses with the listed epigenetic features. Three different comparisons were made: all proviruses versus randomly generated control (*left*) and proviruses from proliferated versus non-proliferated cell clones (*right*). The odds ratios and the *p* values were calculated using the two-sided Fisher's exact test. n.s., *P* > 0.05. **b** Log$_{10}$ transformed chromosomal distance (bp) to the nearest histone modification among proviruses from proliferated versus non-proliferated cell clones. *n* represents the number of proviruses. **c**–**e** Log$_{10}$ transformed chromosomal distance (bp) to the nearest transcription start site (TSS) (**c**), same-direction TSS (**d**), and ATAC-peak among proviruses (**e**) among proviruses from proliferated versus non-proliferated cell clones. *n* represents the total number of proviruses (**c**, **e**) or number of same-orientation proviruses (**d**). Horizontal bars represent mean ± SD. n.s., *p* > 0.05. *P* values were calculated using the two-tailed Mann-Whitney test (**b**–**e**). Source Data Fig. **a**–**e**.

is associated with facultative heterochromatin, indicating repressed transcriptional activity. These results suggest there could be subsets of proviruses with distinct epigenetic marks.

### Proviruses in proliferated cell clones align to chromatin regions favoring transcriptional activation

Due to limited sensitivity of previous assays, it has been difficult to compare epigenetic patterns of proviruses from proliferated versus non-proliferated cell clones. Here we examined whether proliferated cell clones might demonstrate genetic or epigenetic features indicating their association with virus production and viremia. Using the method of integration site analysis above, aligned integration sties to the human genome and found proviruses in proliferated versus non-proliferated clones were 1.1-fold more likely to of be in genes (Fig. 6a, *right column*). Moreover, upon alignment of the integration sites to the ENCODE CHIP-seq data set on resting primary CD4 + T cells, we found proviruses in proliferated cell clones were more likely to align to regions enriched in activating histone modifications such as H3Kme3, H3K27ac, H3K4me1, and H3K36me3 and less likely to be aligned to regions of heterochromatin associated with H3K9me3 (Fig. 6a, *right column*). Among the proviruses that aligned outside histone

modification regions, we found integration sites in the proliferating cell clones aligned closer to regions associated with activating histone modifications and further away from regions associated with the inhibitory histone modification H3K9me3 (Fig. 6b). We also found integration sites from the proliferated cell clones mapped significantly closer in chromosomal distance to transcription start sites (TSS), same-orientation TSS, and ATAC-peaks compared to integration sites from non-proliferated cell clones (Fig. 6c-e). These results indicate proviruses in proliferating cell clones aligned to genetic and epigenetic regions in the host chromosome associated with proviral transcription, which could reflect that proliferated cells have more accessible regions for HIV integration to occur. Importantly, these results indicate some proviruses in proliferated cell clones with activating epigenetic marks were not extensively eliminated and could be detected in this study.

### Proviruses in proliferated cell clones associated with viremia do not align to chromatin regions favoring transcriptional repression

Next, we questioned whether proviruses associated with viremia were preferentially eliminated. We found a similar distribution in genes and

genome regulatory elements among proviruses associated with or without viremia (Fig. 7a, *left column*). However, proviruses associated with viremia aligned to chromatin regions that significantly disfavored or were further away in chromosomal distance to all the activating histone modifications H3K4me3, H3K27ac, H3K4me1 and H3K36me3 compared to proviruses not associated with viremia (Fig. 7a, *left column*, b). In addition, proviruses associated with viremia aligned to chromatin regions were significantly further away in chromosomal distance to same-orientation TSS, but similar in distance to the nearest TSS and ATAC-peak (Fig. 7c). These results suggest a selection process, in which the proviruses associated with viremia that had activating chromatin features were extensively eliminated, leaving behind proviruses with more repressive chromatin features.

Among the subset of proliferated cell clones, we found proviruses associated with and without viremia were in similar distribution patterns in genome regulatory elements and alignment (Fig. 7a, *right column*). In addition, proviruses associated with viremia were not significantly disfavored for the activating histone modifications H3K27ac, H3K4me1 and H3K36me3 compared to proviruses not associated with viremia. Indeed, among proliferated cell clones, proviruses associated with viremia were only significantly further away in chromosomal distance to one activating histone mark H3K36me3 (Fig. 7d), which was in contrast to all proviruses associated with and without viremia (Fig. 7b). Also, among proliferated cell clones, proviruses associated with viremia were not significantly further away in chromosomal distance to the nearest same-orientation TSS (Fig. 7e), which was in contrast to all proviruses associated with and without viremia (Fig. 7c). Collectively, among the proliferated cell clones, proviruses associated with and without viremia demonstrated similar activating chromatin features, which suggested their contribution to virus production. Further, preferential elimination of proviruses associated with viremia was less prominent among proliferated cell clones.

### Proviruses in proliferated cell clones have persistent activating chromatin features over time

To assess whether proliferated cell clones exhibited preferential elimination over time, we analyzed the chromatin landscape of proviruses in proliferated and non-proliferated cell clones during early infection, ART suppression, and rebound infection. Proviruses in proliferated versus non-proliferated cell clones were consistently more likely to be distributed in genes and were more likely to be in enhancers during rebound infection (Fig. S11a). We found the proviruses in proliferated cell clones were more likely to align to regions enriched in activating histone modifications H3K27ac, H3K4me1, and H3K36me3 during early infection (Fig. S11a, *left column*). Interestingly, during ART suppression, proviruses in proliferated cell clones no longer favored alignment to regions enriched in activating histone modifications H3K27ac and H3K4me1 compared to non-proliferated cell clones (Fig. S11a, *middle column*). During rebound infection, proviruses in proliferated cell clones were more likely to align again to regions enriched for two (H3K4me1 and H3K6me3) out of the three histone marks favored during early infection (H3K27ac, H3K4me1, and H3K36me3) (Fig. S11a, *right column*). For proviruses in proliferating cell clones that aligned to regions that were not enriched in histone modifications, the chromosomal distance to the nearest activating histone marks significantly increased between early and rebound infections (Fig. S11b), suggesting rebound virus and antigen production re-initiated elimination of transcriptionally active proviruses. Among proviruses in proliferated cell clones we did not observe significant differences in their alignment to the nearest TSS, same-orientation TSS, and ATAC-peaks over time (Fig. S11c-e). These results suggest during ART suppression, proviruses among proliferated cell were positively associated with genes, promoters, and one activating histone medication H3K36me3, but demonstrated a loss of association with some of the positive

transcription marks (Fig. S11b). Thus, we suspect some preferential elimination of transcriptionally active proviruses occurred among proliferated cell clones over time, but could not effectively out-compete against cellular clonal proliferation.

## Discussion

This study presented a sophisticated series of sensitive and high-throughput PCR-based methods combined with deep sequencing to measure a multi-dimensional dataset of HIV-1 barcoded proviral lineages and integration sites simultaneously with viral RNA lineages at a single molecule level in humanized mice infected with barcoded HIV-1. By incorporating UMI to each viral RNA and proviral DNA molecule in parallel, we quantified the number of viral RNA and proviral barcodes and proliferated cells, which was more accurate and had a wider range of quantification than previous method using the length of DNA ends from random fragmentation[52]. In addition, the sensitivity of this platform allowed us to retrieve 890 viral RNA barcodes and 504 proviral barcodes linked to 15,305 integration sites. Because we measured viral barcodes from the extracted RNA of virus in the plasma, we could use BI-seq to classify which proviral barcodes were associated with viremia. This detailed level of information allowed us to make the following observations.

To begin with, the barcoded virus technology allowed us to accurately track the original input viral barcodes or lineages and their seeding capabilities over time without requiring deconvolution of viral lineage evolutionary trees. The BI-seq platform accurately measured minor viral barcodes in low abundance, which was important in assessing the genetic diversity of all the proviral barcodes. Recently work suggests the genetic diversity of proviral lineages is stable despite cellular clonal proliferation during ART suppression[35]. However, previous studies have not used integration site analysis to analyze the effect of cell clonal expansion on the genetic diversity of proviral lineages throughout early infection, ART suppression, and rebound infection after ART interruption. Even though we detected cell clonal proliferation during early infection, ART suppression, and rebound infection, it did not significantly alter the genetic diversity of proviral lineages. Importantly, the genetic diversity of proviruses remained stable even during rebound infection following ART interruption when dominant circulating viral barcodes seeded as proviral DNA. To explain this, recent work with barcoded SIV has found that individual rebounding lineages were usually the predominant one right before ART was initiated, thus their potential to reseed the reservoir after ATI was likely overshadowed by their initial pre-ART contribution to the reservoir[53]. The genetic stability of proviral lineages throughout all phases of infection in our study suggests that any intervention targeting the reservoir would need to effectively target a diverse range of proviral variants to achieve successful eradication.

Elucidating the mechanisms behind viral persistence is crucial to devising targeted strategies for reservoir reduction or elimination. Here we found cellular clonal proliferation and viral seeding into proviral DNA were strongly associated with one another particularly during rebound infection after ART interruption, suggesting that proliferated cell clones can contribute to viremia. To further assess whether the proliferated cell clones contributed to viremia, we compared the genetic and epigenetic features of 15,305 integration sites from proliferated versus non-proliferated cell clones. Interestingly, we found proviruses in proliferated cell clones were associated with transcriptional activation compared to non-proliferated cell clones, which was consistent with our hypothesis that proliferated cell clones actively contributed to viremia and viral seeding, potentially fueling viral persistence and rebound infection. In addition, we found proliferated cell clones associated with viremia tended to be smaller in clone size (UMI per IS) compared to proliferated cell clones not associated with viremia. Further, there was a negative correlation between the size of the cell clone and the extent of seeding of its viral barcode (UMI per BC). Thus, these results suggest

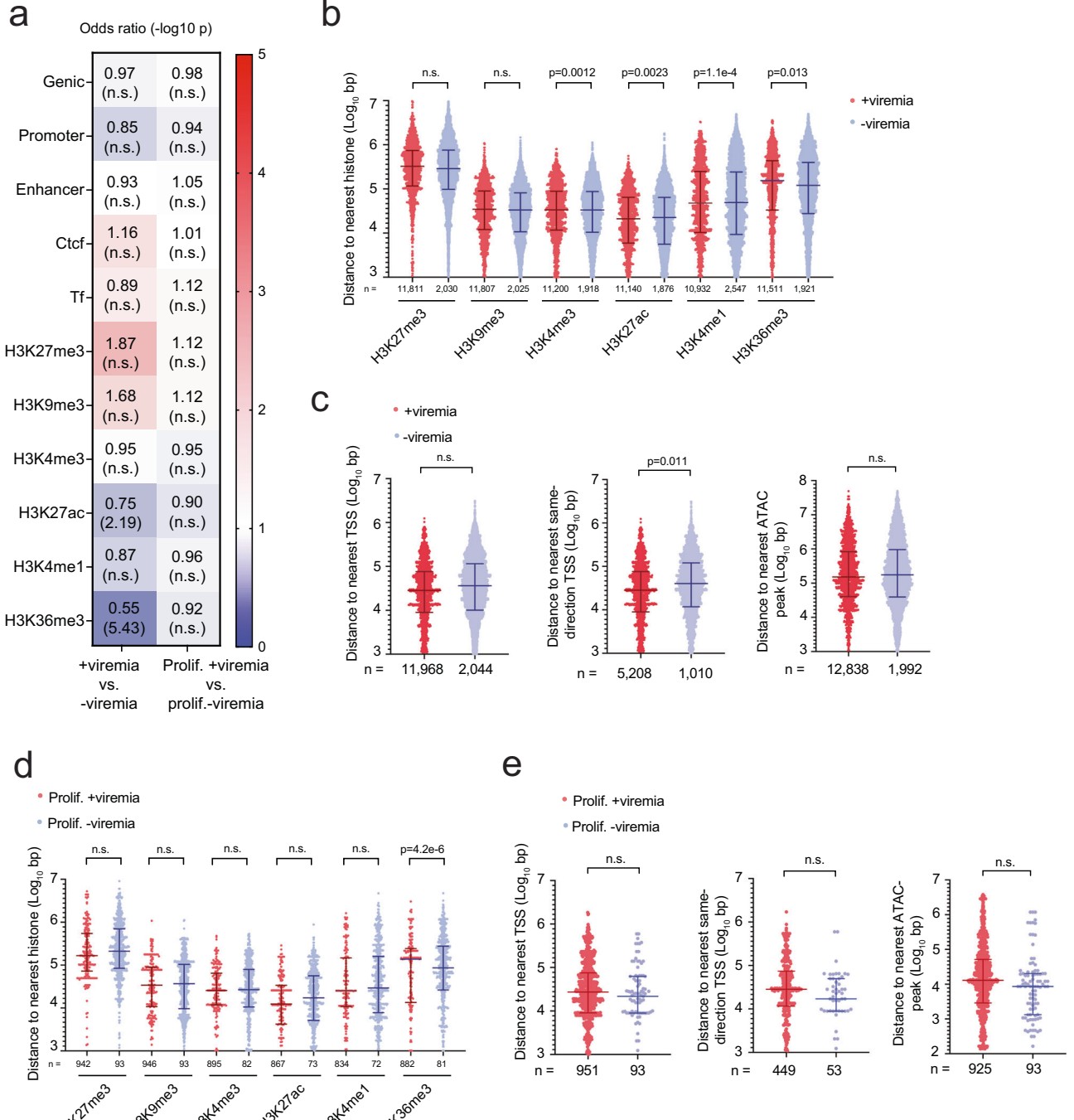

**Fig. 7 | Proviruses associated with viremia in proliferated cell clones have activating chromatin features. a** Heat map demonstrating the odds ratio of proviruses with the listed genetic and epigenetic features. Two different comparisons were made: proviruses associated with (+viremia) or without (-viremia) viremia (*left*), proviruses among proliferated cell clones associated with (+viremia) or without (-viremia) viremia (*right*). The odds ratios and the *p* values were calculated using the two-sided Fisher's exact test. n.s., $P > 0.05$. **b** Log$_{10}$ transformed chromosomal distance (bp) to the nearest histone modification among +viremia or -viremia provirses. *n* represents the number of proviruses. **c** Log$_{10}$ transformed chromosomal distance (bp) to the nearest transcription start site (TSS) (*left*), same-direction TSS (*middle*) and ATAC-peak (*right*) among +viremia versus -viremia

proviruses. *n* represents the total number of proviruses (*left, right*) or number of same-orientation proviruses (*middle*). **d** Log$_{10}$ transformed chromosomal distance (bp) to the nearest histone modification of +viremia versus -viremia proviruses among proliferated cell clones. *n* represents the number of proviruses among proliferated cell clones. **e** Log$_{10}$ transformed chromosomal distance (bp) to the nearest transcription start site (TSS) (*left*), same-direction TSS (*middle*) and ATAC-peak (*right*) among +viremia versus -viremia proviruses in proliferated cell clones. *n* represents the number of proviruses among proliferated cell clones. Horizontal bars represent mean ± SD. n.s., $p > 0.05$. P values were calculated using the two-tailed Mann-Whitney test (**b**–**e**). Source Data **a**–**e**.

proliferated, but not massively expanded cell clones, were important to viremia including rebound viremia and viral spread.

It has not been clear how proliferated cell clones can withstand elimination forces triggered by proviral transcription or virus

production. Prior studies indicate that proviral transcription should trigger preferential elimination and drive the proviral reservoir into deeper latency over time[29,54]. Recent studies analyzing the chromatin landscape of proviruses in elite controllers and PLWH on long-term

ART suppression have shown longitudinal accumulation of proviruses in heterochromatin[12,13]. Recent work has also suggested infected cell clones harboring intact or defective proviruses can spontaneously express viral RNA and protein during ART suppression and elicit HIV antigen-specific CD4+ and CD8 + T cell responses[11,39]. In addition, viral transcription can trigger intracellular pathways leading to cell death[55,56]. Thus, the production of viral transcripts, antigens, or viral particles from either intact or defective proviruses can induce host cell death or elimination by the immune system, leaving behind a more transcriptionally silent reservoir. In support of these studies, we found proviruses associated with viremia had significantly less activating epigenetic features compared to proviruses that were not associated with viremia, suggesting that transcriptionally active proviruses were overall preferentially eliminated. It has been demonstrated that BLT mice can elicit broad and highly specific anti-HIV CD8 + T cell responses akin to humans[57]. Additionally, after HIV vaccination, BLT mice can demonstrate modest, but significant HIV gag-specific T cell responses associated with a decline in viral replication in vivo[58]. Thus, it is possible that cytotoxic anti-HIV T cells may exert pressure on transcriptionally active provirus-harboring cells in the BLT mouse model. However, the mechanisms of elimination of transcriptionally active cells in this model will need further evaluation since BLT mice do not have fully functional human immune systems and may not fully recapitulate the reservoir selection process occurring in ART-suppressed individuals.

Despite the apparent elimination of transcriptionally active proviruses in this context, we detected persistent proliferated cell clones harboring proviruses associated with transcriptional activation compared to non-proliferated cell clones during early infection, ART suppression, and rebound infection after ART interruption. In addition, among proliferated cell clones, proviruses associated with viremia did not align to chromatin regions favoring transcriptional repression, indicating proliferated cell clones associated with viremia were persistent. However, we do not claim that proliferated cell clones were entirely resistant to elimination. During ART suppression, proviruses among proliferated cell clones were no longer associated with some of the activating histone marks (H3K27ac and H3K4me1) observed during early infection. Further, during rebound infection, proviruses in proliferated cell clones regained some, but not all, of the activating histone marks that were observed during early infection. Thus, the elimination of transcriptionally active proviruses, which was active even during ART suppression, and possible reduction of proliferation rates among infected cell clones[14] could not outcompete total cellular proliferation of cell clones harboring proviruses. This likely explains why long-term ART suppression cannot eliminate the reservoir and additional therapeutics are needed to target specific persistent cell subsets within proliferated cell clones.

It is also possible that proliferated cell clones may have mechanisms in place to evade targeted elimination. However, we did not find strong evidence that a post-integration selective advantage was conferred to cell clones based on integration into certain genomic loci. Only 1.2% of proliferated cells had proviruses integrated in the same-orientation integration as a gene associated with cellular proliferation and survival (WWC1) and demonstrated a significantly higher cellular proliferative capacity compared to other proliferated cells. Thus, for the majority of proliferated cell clones proviral integration in an oncogene was not the major mechanism of post-integration cellular proliferation or survival. It is possible that prolonged exposure to viral antigens can sometimes induce a state of tolerance in the immune system, where it becomes less responsive to proliferated cell clones harboring transcriptionally active proviruses. It is also possible these proliferated cell clones may be located in immune privileged sites. Targeting the persistence mechanism of proliferated cell clones harboring proviruses could bolster approaches for reducing the size and

transcriptional activity of the viral reservoir, ultimately aiding efforts towards a functional cure for HIV.

There are several limitations to this study. We did not longitudinally track viral barcodes and infected cell clones from mouse blood samples, but given the sensitivity of our platform, this may be possible in future studies. Infected animals were treated with six weeks of ART due to the constraints of the animal model. This may not fully mimic kinetics of PLWH who can be on ART suppression for decades. Nevertheless, even within our time frame, we observed significant dynamic changes in the proviral lineages, some of which are consistent with human studies of the viral reservoir. The barcoded virus technology has not yet been adapted for single molecule full length viral genome sequencing and single cell transcriptomic or proteomic analysis. Further adaptation of the barcoded virus technology will be critical for advancing animal studies of the viral reservoir. Despite these limitations, the study sheds light on the complex dynamics of HIV reservoirs and offers key insights that may inform the development of therapeutic strategies. Future research could focus on addressing these limitations to further refine our understanding of viral persistence and to develop more effective approaches for HIV treatment and cure.

## Methods
### Mice
All mice were maintained in the Biosafety level 2 plus (BSL2 + ) animal facility at UCLA. Humanized bone marrow liver thymus (BLT) mice were constructed by the UCLA humanized mouse core using conventional methods[59]. In brief, B6.129S-Rag2tm1Fwa Cd47tm1Fpl Il2rgtm1Wjl/J or TKO mice[32] were purchased from The Jackson Laboratory. All mice were housed in individually vented cages in a specific pathogen-free BSL2+ animal facility with a 12 hr light/12 hr dark cycle, between 18-23 °C ambient temperature, 40-60% humidity, and provided food and water in abundance. Male and female mice were age-matched and between 6 and 8 weeks old were irradiated with 270 rads, and then pieces of fetal thymus and liver tissue were transplanted under the kidney capsule. Mice were then injected intravenously with $5 \times 10^4$ human fetal liver-derived CD34+ cells isolated by immunomagnetic separation. Humanized mice infection and sample harvest. All mouse experiments were performed in ethical compliance with the study protocol approved by the UCLA Animal Research Committee (ARC-2021-020).

### Primary cells and cell lines
To isolate primary human CD4[+] T cells, adherent macrophages were removed from PBMCs by culturing in flasks overnight in C10 (RPMI 1640 media supplemented with 10% vol/vol FBS (Omega Scientific, NC2117537), 1% L-glutamine, 1% penicillin/streptomycin (Gibco, 10378016), 500 mM 2-mercaptoethanol (Sigma, M6250), 1 mM sodium pyruvate (Gibco, 11360070), 0.1 mM MEM nonessential amino acids (Gibco, 11-140-050), 10 mM HEPES (Gibco, 15-630-080), and 20 ng per ml of recombinant human interleukin-2 (IL-2) (Peprotech, 10 µg). HEK293T/17 cell line was purchased from the American Type Culture Collection (ATCC, CRL-11268). The following reagent was obtained through the NIH AIDS Reagent Program, Division of AIDS, NIAID, NIH: GHOST (3) CXCR4[+]CCR5[+] cells from Dr. Vineet N. Kalamansi and Dr. Dan R. Littman (cat.# 3942). De-identified PBMCs from healthy human donors were obtained from the UCLA AIDS Institute Virology Core Laboratory and provided to investigators in an anonymized fashion. Informed consent was not required here as it was obtained during original sample collection (not part of this study) and only de-identified samples were used in this study.

### Barcoded HIV-1 library construction
The design of the barcoded virus is shown as a synthetic 21 bp barcode sequence, similar to what we have done previously in another X4-

tropic HIV strain NL43[60], that was cloned downstream of *env* in the full-length R5-tropic HIV-1 NFNSX (Fig. 1a). The molecular barcode sequence was constrained by a cytosine every 3rd nt to avoid introduction of unwanted start or stop codons. A short fragment of the *nef* Kozak sequence was repeated to keep the barcode from affecting *nef* translation. Briefly, two fragments covering upstream and downstream of the barcode region were PCR amplified using following primers: makeBC_F2 and makeBC_R2, makeBC_F1 and makeBC_R1 (Supplementary Data 6). The products were then purified and eluted in TE buffer. The 2 fragments have 20 bp overlapping region, then they were assembled together using NEB HiFi Assembly kit (NEB, E5520). The assembled fragment was amplified again using primers makeBC_F1 and makeBC_R2. The fragment was digested by restriction enzyme NcoI-HF and EcoRI-HF, and purified using PureLink PCR clean-up kit (Invitrogen, K3100). The NFNSX vector was also digested by these 2 enzymes and was purified by agarose gel electrophoresis. One μg of the insert fragment and 5 μg of the vector was assembled using NEB HiFi Assembly kit. The assembled DNA was purified by ethanol precipitation with Pellet Paint NF co-precipitant (EMD millipore, 70748). One μg of the purified DNA was transformed into the Stbl4 *E. coli* electrocompetent cells (Invitrogen, 11635) and plated to 20 15 cm agar plates. The plates were cultured at 37 °C overnight. More than 0.5 million colonies were scratched from the surface of the plates. One mg plasmid DNA was extracted from the bacteria pellet. The barcode region on the plasmid was confirmed by Sanger sequencing and deep sequencing. The library of plasmid DNA was subjected to Illumina deep sequencing and yielded a complexity of 187,530 barcodes with uniform frequency (Fig. S1a). Within the barcode region, there was an average of ~10 nt difference between any two randomly selected barcodes, enabling easy identification of the barcodes using deep sequencing (Data Fig. S1b).

Plasmid DNA was transfected into 293 T cells using following manufacturer's protocol for the BioT transfection reagent (Bioland, B01-01). The virus library was harvested 48 hours after transfection and passed through a 0.45 μm filter. DNaseI (40 ng/mL) was added to the library to remove residual plasmid DNA from the supernatant. The frequency of the barcodes remained identical after virus packaging in HEK293T cells and passaging in costimulated primary human CD4 + T cells (Fig. S1c, d), indicating the library of barcode diversity was maintained during virus packaging and replication in vitro. The resulting NFNSX barcoded virus (NFNSX-BC) supernatant generated by transfected HEK 293T contained similar levels of HIV p24 compared to the parental non-barcoded NFNSX as quantified by ELISA (Fig. S1e). The barcoded virus library was aliquoted and frozen at −80 °C for future use.

### In vitro HIV-1 infections
GHOST (3) CXCR4 + CCR5+ cells were cultured in DMEM containing 10% vol/vol FBS, 500 μg per mL G418 (Gibco, 10131027), 1% penicillin/streptomycin, 100 μg per mL hygromycin (Sigma, 10687010), and 1 μg per mL puromycin (Sigma, A1113802). Cells were seeded into 24-well tissue culture plates at $5 \times 10^4$ cells per well. The next day media was replaced with various doses of HIV-1 and fresh media containing 10 μg per mL of polybrene (Millipore Sigma, TR-1003). Plates were incubated for 2 h at 37 °C and then the media was replaced with 1 mL of fresh media without polybrene. Cells were incubated for a further 2 days, and then harvested by exposure to 0.05% trypsin-EDTA (Gibco, 15400054) in phosphate buffer saline (PBS) (Gibco, 10-010-023) for 5 min, and then removed by agitation with media. Cells were collected and fixed in 3% paraformaldehyde then analyzed for GFP expression by flow cytometry (Fig. S12a). In particular, the NFNSX-BC and parental NFNSX virus were used to infect the reporter cell line GHOST CXCR4+CCR5+ (Fig. S1f), indicating the barcode sequences posed no selection advantage to the virus replication in vitro.

CD4 + T cells were isolated from PBMCs by immunomagnetic selection using CD4 MicroBeads (Miltenyi, 130-097-048) following the manufacturer's instructions, then were co-stimulated with Dynabead CD3/CD28 human T-activator (ThermoFisher Scientific, 11131D) per manufacturer's instructions and cultured in C10 media containing 20 ng per ml of recombinant human IL-2. For infection, $5 \times 10^5$ cells were exposed to HIV-1 in 200 μl of C10 media containing recombinant human IL-2 and 10 μg per mL of polybrene. Cells were spin inoculated by centrifugation at 1200×g for 1.5 h at 22 °C. Cell-free supernatant was harvested 48 hours after transfection and passed through a 0.45 μm filter. DNaseI (40 ng/mL) was added to the library to remove residual DNA from the supernatant. The barcoded virus supernatant was aliquoted and frozen at -80 °C for future use.

### Plasma viral load measurements
For interval biweekly or weekly bleeds, 50 μl of blood was collected using EDTA-coated capillary tubes by retro-orbital bleed for viral load measurements. Whole blood was spun at 300×g for 5 min to separate plasma from the cellular fraction. Total RNA was extracted from plasma using QIAamp Viral RNA Mini Kit per the manufacturer's protocol. HIV-1 RNA was quantified by qRT-PCR. The reaction mixture was prepared using Taqman Fast Virus 1-Step Mastermix (ThermoFisher Scientific, 4444432) with 20 μl eluted RNA and a sequence-specific targeting a conserved region of the HIV-1 gag gene probe, and HIV-1-gag_F1 and HIV-1-gag_R1, respectively (Supplementary Data 6). Cycle threshold values were calibrated using standard samples with known amounts of absolute plasmid DNA copies. The quantitation limit was determined to be 200 copies per ml. At necropsy, more than 100 μl of blood was collected from each mouse, which allowed at least 50 μl of plasma to be analyzed for RNA barcode analysis.

### Blood and tissue harvest and processing
Plasma was separated from blood and then the remaining layer was lysed using RBC Lysis Buffer (Biolegend, 420301) to obtain the PBMCs. Splenocytes and human thymic implants were obtained by passing disaggregated tissue through a 40 μm filter. Bone marrow cells were obtained by grinding bones using a mortar and pestle. Blood, spleen, and bone marrow cells were then fixed for flow cytometry as described below. The remaining cell pellets were stored for DNA analysis or suspended in RLT buffer (Qiagen, 79216) for RNA analysis and frozen at −80 °C.

### RNA barcode analysis
RNA was extracted from plasma using QIAamp Viral RNA Mini Kit (Qiagen, 52904) and from cells using RNeasy Mini Kit (Qiagen, 74104). cDNA was generated using SuperScript IV First-Strand Synthesis Kit (Invitrogen). The same amount of input RNA that was used for viral load measurement was also used for cDNA synthesis for barcode analysis[33]. The barcode region was amplified by hemi-nested PCR using Phusion High-Fidelity DNA Polymerase (ThermoFisher Scientific, F-530XL). Each RNA molecule was tagged with a Primer ID. Primer ID removal and cDNA purification was performed using a Purelink Quick PCR Purification Kit (Invitrogen, K310001). For the first PCR reaction, a fixed volume of 11 μl of cDNA was used along with the following primers (Supplementary Data 6): vRNA-BC_F1 and vRNA-BC_R1. The second PCR reaction used 2 μl of the first PCR reaction, the same reverse primer, and the following forward primer: vRNA-BC_F12. The amplified fragment was ligated with the sequencing adapter using NEBNext UltraII End Repair/dA-Tailing Module (New England Biolabs, E7546L) and NebNext UltraII Ligation Module (New England Biolabs, E7595L). A six-nucleotide multiplexing ID was used to distinguish among different samples. Deep sequencing was performed with Illumina HiSeq3000 PE150. We ensured sequencing depth is tenfold higher than viral genome copies. Raw sequencing reads were de-multiplexed using the six-nucleotide ID. Sequencing error within the barcode region was

corrected by filtering out low-quality reads (quality score <30) and unmatched base pairs between forward and reverse reads. We also used Primer IDs to correct sequencing errors. We filtered the Primer IDs using the frequency cutoff between the Poisson distribution of errors and normal distribution of real Primer IDs[33]. For each Primer ID, the most frequently observed barcode was called. We then grouped the similar barcodes into clusters. Clustered barcodes represent sequences with ≥4 bp differences from one another.

## Staining and analysis for flow cytometry

Cells from animals were stained with fluorescently conjugated antibodies: CD14-Brilliant Violet 510 (clone M5E2), CD3-Pacific Blue (clone Hit3a), CD8a-FITC (clone Hit8a), CD4-PE (clone RPA-T4), CD19-PE-Cy5 (clone SJ25C1), CD45-APC (clone 2D1), CD56-PE-Cy7 (clone MEM-188) (all from Biolegend) and Ghost Dye Red 780 (Tonbo Biosciences). All flow cytometry samples were run using a MACSQuant Analyzer 10 flow cytometer (Miltenyi) or Attune NxT (Beckman Coulter). All data was analyzed using FlowJo v.10.6.0 (TreeStart, Inc) (Fig. S12).

## Cell-associated (CA)-HIV RNA and total HIV DNA

Cell-associated HIV RNA was extracted from lysed splenocytes, bone marrow cells, and human thymic implants using RNeasy Mini Kit. Genomic DNA was extracted from cell pellets using DNeasy Blood & Tissue Kits (Qiagen, 69504). Viral loads were measured by qPCR using 500 ng of CA-HIV RNA and DNA with the same primers as above (Supplementary Data 6). The same amount of input RNA and DNA that was used for viral load measurement was used for barcode analysis.

## DNA sequencing library preparation

Building upon previous methods using inverse PCR to map proviruses[60,61], the barcode and integration site linkage sequencing library was prepared as the workflow shown in Fig. S3b with primers in Supplementary Data S6. Mouse DNA and RNA was extracted using Allprep DNA/RNA Mini kit (Qiagen, 80204). One μg DNA was subject to enzymatic fragmentation using HinP1I (NEB, R0124S). The digested DNA was then purified by PureLink PCR clean-up kit. UltraII End-repair Module (NEB, E7595) prepared the DNA for ligation. A custom adaptor was annealed in the TE buffer. The sequence of the adapter's reverse strand is TTGAGGTTTGCAGTTG with a 5' modification of a phosphorylation group to facilitate TA ligation with the genome fragments. The 3' amino modification blocks the polymerase from adding nucleotides at its downstream, maintaining the L-shape conformation of the adapter. Three consecutive phosphorothioate bonds at the 3' end stabilize the adapter, preventing it from enzymatic degradation. The forward strand of the adapter is ACCATCAACCCC-GAATTCNNNNNNNNNNNNNNNCAACTGCAAACCTCAAT. It anneals with the reverse strand and contains a 14-nucleotide UMI. 50 pmol of adapter was ligated to 1 ng of fragmented genomic DNA. All ligated product was purified and amplified using 4 rounds of semi-nested PCR. All PCRs used the same reverse primer DNA-BC_R1. Previous barcode-integration site linkage[60] was modified so that the forward primer sequences annealed to different parts of HIV-1 genome to increase the PCR specificity. They are in the order of DNA-BC_F4, DNA-BC_F3, DNA-BC_F2 and DNA-BC_F1. All forward primers contain three consecutive phosphorothioate bonds at the 3' end, preventing the exonuclease activity of the polymerase, increasing the PCR specificity. Primers F3 and F2 were used with the reverse primer containing the 5' phosphorylation modification, to enable lambda exonuclease digestion after PCR, which can eliminate the product of unspecific amplification. The final PCR product was purified and digested by EcoRI-HF (NEB, R3101S). Ten ng DNA was purified and subjected to self-ligation in a 100 μL reaction. The reaction used 2 units of T4 ligase (Invitrogen, 15224017) in room temperature for 4 h. The intermolecular ligation efficiency was confirmed by quantitative PCR using primers DNA-BC_ivF and DNA-BC_ivR and synthetic standard templates. The intermolecular ligation efficiency was also assessed using quantitative PCR with only DNA-BC_ivR primers. One third of the ligation product was used as the PCR template for the inverse PCR, using the same primers as the quantitative PCR. Phosphorothioate bond modification was used to increase PCR specificity. One tenth of the product was then subject to the final round of PCR, which adds Illumina sequencing adapters to the library. The primers are Illumin_F1 and Illumina_R1. N stands for indexing sequence to distinguish different samples. The final product was confirmed using gel electrophoresis and purified by the PCR clean-up kit. We then use NEBNext Ultra II DNA library prep kit (New England Biolabs, E7103) to make pair-end sequencing libraries. All libraries were mixed and purified for Illumina NovaSeq6000 PE150 sequencing. Ten million reads were retrieved for each sample.

## Barcode diversity analysis

In addition to counting the total number of barcodes detected in each mouse per organ, two additional measures of diversity of barcodes have been used. Denote by $N$ the total number of distinct barcodes in an organ, by $n_i$ the count of barcode $i$, with $i = 1,...,N$, and $n = \sum_{i=1}^{N} n_i$. The frequency of barcode $i$ is simply $p_i = \frac{n_i}{n}$. Assume that the list is ordered with index 1 referring to the most frequent barcode: $n_1 \geq n_2 \geq \ldots \geq n_N$. Then Shannon diversity index was calculated as $-\sum_{i=1}^{N} n_i \ln p_i$. Further, the Simpson dominance index is given by $\sum_{i=1}^{N} \frac{n_i(n_i-1)}{n(n-1)}$.

## DNA barcode and integration site data analysis

The data analysis pipeline was summarized in Fig. 2. For the barcode and integration site linkage sequencing, the barcode and UMI was extracted by mapping their flanking sequences. The sequence downstream of the HIV-1 LTR and upstream of the L-shape adapter was extracted as provirus integration site. If the sequence mapped to the plasmid of NFNSX, it was discarded as contamination. If the sequence was less than 10 nucleotides, it was considered as linear unintegrated provirus. The rest of the sequences were aligned with human genome hg38 Ensemble release 108 or the NFNSX genome by bowtie2[45], and classified as integrated or auto-integrated. If the sequence mapped immediately downstream of the HIV-1 LTR, it was classified as circular. We then identified the true UMIs from the sequencing errors by counting the occurrence of the UMIs. The count of the true UMI should follow a normal distribution while the sequencing errors were Poisson distributed. We set the threshold of calling true UMI for each sample to the separation point of the bi-modal count distribution. Then we assigned the most commonly observed barcode and integration site for each UMI. With the help of UMI, we identified the barcode and integration site for each provirus molecule. The barcode and UMI in RNA samples were also retrieved by mapping their flanking sequences. The true UMI was identified by its count distribution. The occurrence of barcode in RNA was quantified by counting UMI. An extra clustering step was carried out for barcode to reduce sequencing errors. For Custom codes for mapping and counting were available upon requests.

## Statistical analysis

Data are presented as pie charts and scatter plots with individual values using Prism 10 version 10.4.0 software (Graphpad). Differences were test for statistical significance by Mann Whitney U tests and Fishers' exact tests as appropriate. $P < 0.05$ was considered significant. Statistical details can be found in the corresponding figure legends. Flow cytometry analyses were performed using FlowJo 10.6.0. Deep sequencing analysis was performed using Bowtie2[45] (v2.4.2), Jupyter (v7.2.1), Bedtools (v2.31.1), SciPy (v1.13.1), and BioPython (v1.83).

## Reporting summary

Further information on research design is available in the Nature Portfolio Reporting Summary linked to this article.

## Data availability

Raw sequencing reads have been deposited at the NCBI Sequence Read Archive (SRA) with the accession code PRJNA963079. Source data are provided with this paper.

## Code availability

All original codes have been downloaded onto github https://github.com/Tian-hao/Barcode-HIV-reveals-viral-persistence[61].

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

## Acknowledgements

We acknowledge support from the National Institutes of Health (AI155232 to J.T.K. and R01AI127410 and R01AI161803 to J.A.Z.), the California HIV/AIDS Research Program (H24BD7864 to J.T.K.), National Center for Advancing Translational Sciences UCLA CTSI Grant (UL1TR001881 to J.T.K.), National Science Foundation (DMS-2152155 to N.L.K and D.W.), and the UCLA Center for AIDS Research (AI152501). This research was supported in part by the CRISPR for Cure Martin Delaney Collaboratory for HIV cure UM1 AI164568-01 and cofunded by NIAID, NIMH, NIDA, NINDS, NIDDK, and NHLBI. The UCLA AIDS Institute, the McCarthy Family Foundation, and the James B. Pendleton Charitable Trust also provided support. ART was generously provided by Merck (RAL) and Gilead Sciences (TDF and FTC). Part of the barcode method development is described in Tian-hao Zhang's PhD thesis at UCLA. Funding agencies did not play a specific role in the conception, design, data collection, analysis, the decision to publish, or preparation of this manuscript.

## Author contributions

Data acquisition, T.Z., Y.S., M.K., A.G. I.A., H.C., G.B., M.D., W.H., C.O., C.C., Y.D., J.T.K., barcode method design, T.Z., R.S., J.T.K.; barcode method development, T.Z., Y.D., R.S., J.T.K.; in vitro and in vivo experimental design and development, J.A.Z, J.T.K, bioinformatics assays and analysis, T.Z., Y.S., J.T.K.; mathematical modeling, N.L.K., D.W.; data interpretation, analysis, and presentation, T.Z., Y.S., N.L.K., D.W., J.T.K.; preparation and writing of manuscript, T.Z., Y.S., N.L.K., D.W., J.A.Z., J.T.K.; discussion, critical review of manuscript, T.Z., Y.S., N.L.K., D.W., C.S., R.S., J.A.Z., and J.T.K.; research idea and concept, T.Z., Y.S., R.S., J.A.Z., and J.T.K.

## Competing interests

J.A.Z. is on the scientific advisory board for BryoLogx and is a co-founder of CDR3 Therapeutics. The other authors declare no competing interests.
