## [Transparent Peer Review file · Nature Communications]

Barcoded HIV-1 reveals viral persistence driven by clonal proliferation and distinct epigenetic patterns

Corresponding Author: Dr Jocelyn Kim

Version 0:

Reviewer comments:

Reviewer #1

(Remarks to the Author)

This manuscript by Zhang et al describes a new mouse model to evaluate HIV persistence. This model is based on the use of a barcoded virus that allows tracking each individual virus both as a provirus and as HIV RNA molecule. Using this barcode virus, the follow genetic viral diversity during acute infection, after ART administration and after ART discontinuation. The authors follow cellular clonal proliferation and viral reseeding and associate these with epigenetic marks and integration sites. The work has some innovation regarding the study of HIV latency and this new technology in mouse models but have some shortcomings as presented.

Major concerns

- The authors evaluate the replicative capacity of the barcoded virus but none of the assays used measure actual replication. The first measure viral release after transfection in 293T cells and the second is a short infection time with mostly one round of replication. Further experiments should be conducted to ascertain that the barcoded virus does not have a defect on viral replication.
- The main concern with the manuscript is that the timeline used may have biased most of the results and may not reflect most infections in humans. First, the levels of viremia achieved before ART initiation are lower that those observed in other studies using similar humanized models. Second, the viremia does not lead to CD4 depletion as observed in humans and in other humanized models. Third, after ART administration and before ART interruption, high levels of viremia can still be detected, what could contribute to some of the conclusions as it seems that the time on ART is not sufficient to fully suppress viral replication. Finally, because of the lack of immune pressure, some of the selection processes may be missing. Albeit the find some similitudes with studies in ART-suppressed people living with HIV, those are minor and as such the conclusions may be misleading. Because of the complexity of the study, a more extensive explanation of all the caveats of the study that has been included should be incorporated. I will suggest to emphasize the technology and deemphasize the relevance to the selection process of the latent reservoir, as this model, as presented, may not fully recapitulate them.

Minor concerns

- Line 69: replace “and early infection” to “and early in infection”
- Reference 21 in line 109 should be replaced by the original articles and not a review.

(Remarks on code availability)

Reviewer #2

(Remarks to the Author)

The three main findings of this manuscript are: 1. the proviral population structure remains unchanged despite clonal expansion and replication off ART. 2. the clonal expansion contributed to proviral expansion. 3. in non proliferated cells (or more precisely, in infected cells where clonal expansion is not identified with the limited sampling) there was an apparent elimination of proviruses associated with transcriptional activation and viremia, while persistent clones had epigenetic

markers of activation.

The work is very specific to HIV and limited to the humanized mouse model. The work largely supports published studies of HIV in humans and SIV in nonhuman primates. While I was very interested in the combination of barcodes and integration sites, I don't think there was any novel discovery using this combined approach. I am left struggling with the excitement of linking the integration site and barcode with a study where the new findings are largely obtained with integration site analysis only. How did the barcodes and your new assay contribute to the finding of the study? I was really hoping for an analysis of which barcodes rebounded, what integration sites were those interesting barcodes found in and if they were evidence for rebound of large expanded clones. It seems that this model and technology was particularly amenable to that type of research question.

One concern regarding the conclusions of the manuscript centers on the length of time required to detect changes in the cellular clones. On lines 77-78 of the introduction, the authors note that it is crucial to know if clones are prone to elimination by selective forces. How long would it take for a viral clone population to be eliminated? Does the xenographic nature of the model affect how and if immune responses can clear infected cells. What viral antigens would need to be expressed and presented to provide this killing. Data from lines 80-82 suggest several decades might be required to detect these changes. This study reports the elimination of proviruses associated with transcriptional activation in non-proliferative cell clones. I believe this is the main point of the paper. However, detection of an expanded clone is based entirely on the total number of infected cells examined. The authors claim their approach is more sensitive but there is no data to support that claim and in fact sensitivity levels are based entirely on the number of cells examined and the number of integration sites obtained. That is all. So conclusions based exclusively on whether or not two or more cells were collected and examined versus only one cell are tenuous at best and prone to false negative results. How large would a clonal population need to be in order to be detected in this study? How many infected cells are there? What is the average and distribution of clones within the population? If you can answer these questions, then one can begin to estimate how many integration sites would be required to claim that there are differences between proliferative clones and non-proliferative clones. Right now, there is a blind assertion that if it wasn't detected it was not a proliferative cell clone. At a very minimum, the limitations need to be spelled out and the name "non-proliferative" changed to a more nuanced title of undetected.

Other suggestions include:

The title of the manuscript is very limited to one part of the paper and doesn't even make sense (distinct chromatin patterns is not informative).

Throughout the paper, the use of "reseeding" to describe infection of cells is very confusing. Re seeding is poorly described word that denotes that the reservoir population is infected. But in most of the uses here, it is just describing infection. Line 64: "re seeding of proviruses by de novo infection of target cells and proliferation of cell clones" assumes that these infected cells will be the reservoir. And while the population in general doesn't change, most individual infected cells will die and not be the reservoir. I would highly recommend for readers that you use active replication pre-ART and ATI vs a block in replication during ART.

Abstract: "despite perturbations such cellular clonal proliferation" should be "such as cellular"

line 70: Not clear what in two individuals means. Two studies? Two donors of cells for humanized mice?

line 411: typo: "integration sties to the human genome"

line 696: Should be "Blood and tissue processing"--not just tissues.

line 162: acute infection (six weeks after HIV infection). Six weeks is well past acute infection. I would change that early or primary infection throughout the manuscript.

line 165: "at intervals" isn't informative.

line 168: No data supporting the statement that engraftment was similar between groups.

Line 169: list out organs at first use.

line 171: "approximately 104 copies per ml" I guess this is true after rebound, but not in primary infection. Should be specific here.

line 178: "In addition to tracking each viral lineage by barcode sequencing" prior to this statement, you haven't discussed barcode tracking. Not sure if this paragraph was in a different place originally, but currently out of place.

lines 178-183: Very poor reasoning for changing methodologies. Limiting dilution works on samples with small total cells as well as with large numbers of cells. Not sure what you are trying to convey here. I think you can delete from 178-184.

lines 208-209: "Using BI-seq as a parallel platform, we examined a total of 293,557 viral DNA molecules from extracted genomic DNA of which 7.3% was integrated." I don't understand how so few genomes were integrated. This is way outside

of normal for HIV in humans and SIV in NHPs. Is this common to this humanized mouse model? Please help readers understand this difference.

line 218--"Persistent HIV infection can be driven by viral reseeding" this is not the conventional way we discuss viral replication. This. and the reseeding score is unnecessarily complicated.

line 282: "proliferated hundreds of times" is a huge underestimate. Be specific in the text: "detected hundreds of times" then you can say, which likely represents thousands of expanded cells. Again, if we knew more about the distribution of clonally expanded cells, we could be more specific on what it really means to detect a single clone "hundreds" of times.

line 291: "The decrease in the cell proliferation index during rebound infection". These data and the ensuing rationalization wasn't clear to me. Viral load was very high so lower proliferation index would suggest that replication in non proliferating cells was the reason for high viral load which is not seen in primary infection. So is there a reason for this or not.

lines 411 and 412: proliferated versus non-proliferated clones were 1.1-fold more likely to of be in genes. Figures 6 and 7 show significance with very small mangnitude changes. I am not a statistician, but it does not pass the common sense test.

(Remarks on code availability)

Reviewer #3

(Remarks to the Author)

In the manuscript entitled "Barcoded HIV-1 reveals proviruses associated with cell clonal proliferation or viremia have distinct chromatin patterns" from Zhang T-H et al., the Authors developed an elegant approach characterize the HIV-1 reservoir in a humanized mouse model of HIV1 infection. To this aim the Authors generated a complex library of HIV genomes containing 21 bp barcodes downstream ENV. This HIV library was then used to infect BLT TKO mice which were euthanized for analysis at three time points to perform endpoint analysis: acute infection (six weeks after HIV injection), ART suppression (six weeks after ART treatment), and rebound infection (six weeks after ART interruption). Genomic DNA and RNA (from tissues and blood plasma) from harvested organs were subjected to linker (Linkage) mediated PCR or PCR to retrieve the viral barcodes and virus/host genome junctions (integration sites), or the barcodes resulting from viral transcriptional activity respectively. By this approach it is possible to study the spread of viral genomes (each marked by a single barcode) across different cell clones (each marked by a different integration site), the relative abundance of each clone (by the analysis of unique molecular identifiers introduced during the barcode and integration site retrieval) and what viral genome is expressed (by the analysis of the barcodes retrieved form RNA). Moreover, analysis of the barcodes in the viral RNA from blood plasma allowed to characterize the repertoire of infectious HIV1 genomes in the different conditions. By this approach the Authors show that the diversity of the proviral reservoir is stable during cellular clonal proliferation and viral reseeding after ART interruption. Moreover, and in agreement with previous studies, the clonal expansion of infected clones appears to be a major source of HIV infection allowing the maintenance of the viral reservoir, viremia and viral reseeding, thus fueling viral persistence. Association of viral integration sites and active or repressive chromatin features varied in acute infection, under ART and rebound conditions and in a way suggesting that the expression of viral genes is counter selected by toxicity or the action of the immune system when under ART. During acute infection however the association with active chromatin was more evident in proliferating cells clones. Although at low frequency the Authors found that some clones with integrations near oncogenes were expanded.

Main comments

This is an interesting paper which describes the development of a new and highly precise strategy to study the HIV1 reservoir in a humanized mouse model. The Authors have confirmed previous findings from other studies in which the clonal expansion of HIV infected clones and, importantly, show a complex interplay between integration in chromatin active or active genomic regions leading to different expression levels of viral genes, counter-selection caused by toxicity of viral infection or immunological counter section but in different contexts selection of proliferating infectious clones during the acute infection. Overall, this is a well written manuscript with interesting findings on HIV persistence in vivo and the factors impacting on reseeding and genomic integration during acute infection, under ART and after ART interruption. Importantly this experimental methodology has the potential to expand our knowledge on HIV1 biology, study the effects of alternative ART or additional/alternative pharmacological interventions aimed at the permanent HIV eradication.

Few minor suggestions are:

1) The strategy to analyze the HIV repertoire by the study of barcodes and integration sites from DNA and barcodes on viral RNA in vivo is quite complex for the non-aficionados. The experimental strategy should be explained in a graphical way

expanding Figure 1 A. I believe that the inclusion of a graphical representation of the experimental strategy would be helpful for the readers.

2) During Illumina sequencing it is possible to find some barcoded oligonucleotides swapped across samples. This phenomenon, which occurs during bridge PCR, does not pose great issues for the sample recognition as wrong sample barcode combinations are easily eliminated bioinformatically. However, this phenomenon could artificially inflate the clonal abundances of specific integrations because different UMIs could be included by the swapping phenomenon. In this case the analysis of the coherence of samples barcodes and integration site will not help. Did the Authors measure the extent of this phenomenon in their datasets? And how to deal with such issue (if present)? In the field on integration site analysis, shear site statistics and statistical modelling algorithms such as "soniclength" (Berry, C. C. et al. Estimating Abundances of Retroviral Insertion Sites from DNA Fragment Length Data. Bioinformatics 2012) have been used with success. It would be interesting if the two methods could be compared.

3) The finding that some integrations in expanded clones targeted oncogenes is interesting and it should be expanded in the discussion. Indeed, although these integrations were rare, they are probably also relevant especially considering the finding that the expanded clones have an important role in the maintenance of the complexity of the clonal reservoir. On this regard it is interesting to note that the Authors did not find enrichment of integration sites targeting BACH2 nor STAT5B. Since these integrations appeared to be enriched in long lived and fully immunosuppressive T-regulatory cells it may be difficult to identify them in this mouse model. Could the Authors elaborate on this aspect? Do these humanized mice produce a detectable number of T-regulatory cells?

(Remarks on code availability)

Version 1:

Reviewer comments:

Reviewer #1

(Remarks to the Author)

The authors have addressed my previous concerns

(Remarks on code availability)

Reviewer #2

(Remarks to the Author)

Congratulations on a comprehensive and highly useful update to your manuscript. I was happy to see the changes that were made and I have no further questions or concerns.

(Remarks on code availability)

Reviewer #3

(Remarks to the Author)

Author's response has addressed my points

(Remarks on code availability)

Dear Editor,

We appreciate the opportunity to respond to the reviewers' comments for our manuscript NCOMMS-24-32622-T to Nature Communications. We appreciate the insightful feedback from the reviewers and the editorial staff, which we hope has been sufficiently addressed in the enclosed point-by-point rebuttal to the reviewers' comments and revised manuscript. The reviewer comments are in blue below. Also, we have included new data in the uploaded revised manuscript to address the reviewers' concerns. Please note the line references are based on when the manuscript is viewed as "simple markup" on Microsoft Word.

Reviewer 1.

The first measures viral release after transfection in 293T cells and the second is a short infection time with mostly one round of replication. Further experiments should be conducted to ascertain that the barcoded virus does not have a defect on viral replication.

We appreciate the comments from Reviewer 1. To assess whether viral production from infected cells was comparable between the two virus isolates in vitro and in vivo, first we conducted infections of GHOST (3) CXCR4+CCR5+ cells with equivalent doses of each virus for 2 hours, followed by two washes. After 48 hours of culture, we measured p24 levels in the cell-free supernatant by ELISA and normalized the level of p24 to the number of GFP+ infected cells per well by flow cytometry (**Fig. S1g**, shown here but also included in the revised Supplementary Information). These results suggest that viral production per infected cell is similar for NFNSX-BC and NFNSX in vitro.

Next, to assess whether viral replication was similar in vivo, we injected an equivalent dose of NFNSX-BC and non-barcoded NFNSX into humanized mice and observed similar viral loads four weeks post-HIV injection (**Fig. S1h**, shown here but also included in the revised Supplementary Information), indicating similar in vivo replication kinetics of the barcoded and non-barcoded HIV strains. We have incorporated these findings into lines 156-165 of the revised manuscript.

Fig. S1: Viral replication in vitro and in vivo.

g, Production of HIV p24 from infected GHOST (3) CXCR4+CCR5+ cells. Equivalent dose of NFNSX-BC or NFNSX was used to infect GHOST (3) CXCR4+CCR5+ cells for 2 h. Infected cells were washed twice and then cell-free supernatant collected at 48h. HIV p24 levels were measured over the number of infected GFP+ GHOST (3) CXCR4+CCR5+ cells. Data is mean \pm SD. $n = 3$ technical replicates per group. P value was calculated using the two-tailed Mann-Whitney test. **h**, Humanized mice were injected with 500ng p24 of NFNSX-BC or non-barcoded NFNSX. Plasma HIV RNA loads were measured by RT-PCR 2 weeks after HIV injection. Data is mean \pm SD. $n = 3$ per NFNSX-BC group. $n = 4$ per non-barcoded NFNSX group. P value was calculated using the two-tailed Mann-Whitney test.

The main concern with the manuscript is that the timeline used may have biased most of the results and may not reflect most infections in humans. First, the levels of viremia achieved before ART initiation are lower than those observed in other studies using similar humanized models.

We appreciate the reviewer's concern and agree there is variability in viral loads across laboratories, which we believe may be due to the different viral load assays. For instance, in our viral load assay we used 1-step RT-PCR with a gene-specific primer as opposed to a 2-step RT-PCR with random hexamers, which is often used by other groups. We have found the 2-step random hexamer RT-PCR yields a 12-fold higher HIV RNA copy count compared to our 1-step gene-specific RT-qPCR with equivalent RNA input (**Rebuttal Fig. 1 as seen here**). This could be due to multiple random hexamers annealing to one RNA molecule during strand displacement allowing for multiple cDNA fragments to be generated from one strand of RNA and increase in PCR efficiency with 2-step vs 1-step RT-PCR. Although we found the hexamer-based RT-PCR protocol yields a higher HIV RNA copy count, we utilized the gene-specific RT primer in our viral RNA RT-PCR and viral RNA barcoding PrimerID assays because it is important to ensure a 1:1 conversion of HIV RNA to HIV cDNA for accurate UMI quantification of the HIV RNA barcodes. We also used 1-step RT-PCR in our viral load assay to decrease the likelihood of PCR contamination between samples, but can consider 2-step RT-PCR in future studies.

Rebuttal Fig. 1: 2-step RT-PCR with random hexamer yields ~12 fold higher copy count compared to 1-step RT-PCR with gene-specific primer. HIV RNA copy count for 1ul of HIV RNA was quantified by either 2-step RT-PCR with random hexamers or 1-step gene-specific primer RT-PCR. Technical replicates are shown. *P* value was calculated using the two-tailed Mann-Whitney test.

Second, the viremia does not lead to CD4 depletion as observed in humans and in other humanized models.

We apologize for the confusion on the previous figure in the original manuscript, which only showed the mean ± SD of the CD4+ T cell frequencies of each animal group. The “ns” (not significant) labels in the previous Fig. S2a meant there were no significant differences between groups that were assigned to be sacrificed or continued to proceed in each timepoint. It was not a statistical test of CD4+ T cell count between different timepoints. Here we now include the CD4+ T cell frequencies of each animal as well as statistical test between different timepoints to clearly show the significant decline in the frequency of CD4+ T cells after 6 weeks of primary HIV infection, which was restored by ART suppression, and then declined again during rebound infection after ART interruption (**Fig. S2a, right** shown here but also included in the revised Supplementary Information). We have revised lines 171-176 of the revised manuscript to include this data.

Fig. S2a, right. Human CD4+ T cell engraftment after NFNSX-BC infection in vivo. Right, Longitudinal frequencies of CD4+ T cells. The *p* values were calculated using paired *t*-test.

We have revised lines 171-176 of the revised manuscript to include this data.

Third, after ART administration and before ART interruption, high levels of viremia can still be detected, what could contribute to some of the conclusions as it seems that the time on ART is not sufficient to fully suppress viral replication.

Overall, viral suppression was achieved for 14 out of 15 infected animals placed on ART. However, there was one animal that had undetectable (<200 HIV RNA copies/ml) after 2 wks of ART, but then detectable viremia (~391 and ~490 HIV RNA copies/ml) at 4 wks and 6wks of ART respectively, but then was undetectable (<200 HIV RNA copies/ml) again 4 days later after ART interruption. The HIV RNA copies detected for this animal while on ART could reflect viral blips or low level viremia. Because the majority of the mice were suppressed on ART, we have shown the median and interquartile range of the viral loads of each group (see **Fig 1c** in the revised manuscript). Indeed, we and others have found ~4-6 weeks of ART typically suppresses plasma viremia in humanized mice, and thus likely indicates latency since rebound viremia occurs a couple weeks after ART interruption²⁻⁷.

Finally, because of the lack of immune pressure, some of the selection processes may be missing. Albeit they find some similitudes with studies in ART-suppressed people living with HIV, those are minor and as such the conclusions may be misleading. Because of the complexity of the study, a more extensive explanation of all the caveats of the study that has been included should be incorporated. I will suggest to emphasize the technology and deemphasize the relevance to the selection process of the latent reservoir, as this model, as presented, may not fully recapitulate them.

We appreciate the reviewer's insight and agree that most infected cells expressing viral antigens will be eliminated either by viral cytopathic effect and/or or host immune clearance. We also agree the host immune clearance of transcriptionally active cells may be less robust in the humanized BLT mouse model. However, it has been demonstrated that BLT mice can elicit broad and highly specific anti-HIV CD8+ T cell responses akin to humans⁸. Additionally, after HIV vaccination, BLT mice can demonstrate modest, but significant HIV gag-specific T cell responses associated with decline in viral replication in vivo⁹. Thus, it is possible that there may be some cytotoxic anti-HIV T cells that exert pressure on transcriptionally active provirus-harboring cells in the BLT mouse model. We agree the mechanisms of elimination of transcriptionally active cells will need further evaluation since humanized mice do not have fully functional human immune systems and may not fully recapitulate the reservoir selection process occurring in people with HIV on ART (lines 561-564).

Minor concerns

- Line 69: replace "and early infection" to "and early in infection."

Thank you, we have corrected this as suggested in the revised manuscript.

- Reference 21 in line 109 should be replaced by the original articles and not a review.

Thank you, we have added the original articles with references 22, 24, and 25 to the revised manuscript.

Reviewer 2

While I was very interested in the combination of barcodes and integration sites, I don't think there was any novel discovery using this combined approach. I am left struggling with the excitement of linking the integration site and barcode with a study where the new findings are largely obtained with integration site analysis only. How did the barcodes and your new assay contribute to the finding of the study?

We thank Reviewer 2 for their comments. We believe the novelty of using the combined approach barcode and linked integration site approach is in its ability to measure integration sites per barcode (IS per BC) as a measure of viral spreading for each viral lineage, which has not been previously reported. Additionally, by linking the barcode and integration site, we were able to differentiate the expanded or proliferated cell clones that were associated and not associated with viremia. Thus, not only could we compare proliferated versus non-proliferated clones, which only requires integration site analysis, but we could also compare proliferated clones associated with or without viremia, which required linkage of proviral barcodes and integration sites.

I was really hoping for an analysis of which barcodes rebounded, what integration sites were those interesting barcodes found in and if they was evidence for rebound of large, expanded clones. It seems that this model and technology was particularly amenable to that type of research question.

We appreciate the reviewer's comments and have provided the following data. We analyzed proviruses associated with rebound viremia and compared integration sites among 834 proliferated vs 11,194 non-proliferated cell clones. Only a small number of integration sites (n=15 or 1.8%) were significantly enriched among proliferated cell clones associated with rebound viremia (**Rebuttal Fig. 2a**). Of these 15 integration sites, five (NBPF12, WWC1, FOXP4, TNFRSF10D, TNFRSF10B) had reported associations with cellular proliferation, survival, or activation, but did not demonstrate higher rates of cellular proliferation compared to the other

Rebuttal Fig. 2: Only a small number of integration sites are enriched proviruses associated with rebound viremia.

a, Dot plot representing the integration events that were enriched in genes among proviruses associated with rebound viremia that were in proliferated versus non-proliferated cell clones. Odds ratio and p values were calculated by fisher exact test. Y-axis indicates the p value ($-\log_{10}$). X-axis indicates the chromosome number and position. Red dashed line indicates the threshold for statistical significance after Bonferroni correction of p-value at 0.05.

b, The 15 integration sites were classified as a known or not known association with cellular proliferation, survival, or activation. The cell clonal proliferation index (UMI per IS) was compared. Horizontal bars represent the median. P value was determined by Mann-Whitney test.

integration sites (NRDC, NBF25P, chr5:155246508, GUCY2EP, chr14:99345793, ENSG00000261520, KCNJ18, chr17:26885518, chr17:82282778, and chr20L5809836). Further, among the genes associated with tumor progression, cell proliferation and anti-apoptosis, only two (WWC1 and TNFRSF10D) were associated with significantly higher cell clonal proliferation indexes (**Rebuttal Fig. 2b**). Collectively, these results indicate integration into genes associated with cellular survival or proliferation was probably not the mechanism of survival or proliferation post-integration for the majority of the proliferated cell clones in our experiment.

We agree with the reviewer that we did find evidence of largely expanded or proliferated cell clones. Further, we found an important association between proliferated cell clones and viremia. Indeed, proliferated cell clones were >19 times more likely to be associated with rebound viremia compared to non-proliferated cell clones (**Fig. 4c**). However, proliferated cell clones associated with viremia tended to be smaller in clone size compared to proliferated cell clones not associated with viremia (**Fig. 4d**). Furthermore, we found a negative correlation between the size of the cell clone (UMI per IS) and the extent of viral spread (UMI per BC) (**Fig. 4e**). Thus, among the proliferated cell clones, we did not find that massively expanded cell clones were more likely to be associated with viremia and viral spread. We have revised lines 331-333 and 536-541 of the revised manuscript to help clarify these findings.

One concern regarding the conclusions of the manuscript center on the length of time required to detect changes in the cellular clones. On lines 77-78 of the introduction, the authors note that it is crucial to know if clones are prone to elimination by selective forces. How long would it take for a viral clone population to be eliminated? Does the xenographic nature of the model affect how and if immune responses can clear infected cells. What viral antigens would need to be expressed and presented to provide this killing. Data from lines 80-82 suggest several decades might be required to detect these changes.

In our experimental model, we were able to detect significant differences in proviral epigenetic marks occurred during each phase of infection. During active viral replication (e.g. primary or rebound infections), cell clones and their proviruses may be rapidly eliminated due to viral cytotoxic effects (and/or immune selection) associated with active virus production. However, during ART suppression, the elimination of cellular clones harboring provirus may be dependent on the rate of spontaneous reactivation of latent clones. As the reviewer astutely suggested, it is possible that xenographic forces may increase the rate of spontaneous reactivation of latent cell clones in ART-suppressed humanized mice compared to ART-suppressed people. We also agree with that it will be important to correlate our findings with ART-suppressed clinical samples in future studies.

This study reports the elimination of proviruses associated with transcriptional activation in non-proliferative cell clones. I believe this is the main point of the paper. However, detection of an expanded clone is based entirely on the total number of infected cells examined. The authors claim their approach is more sensitive but there is no data to support that claim and in fact sensitivity levels are based entirely on the number of cells examined and the number of integration sites obtained. That is all. So conclusions based exclusively on whether or not two or more cells were collected and examined versus only one cell are tenuous at best and prone to false negative

results. How large would a clonal population need to be in order to be detected in this study? How many infected cells are there? What is the average and distribution of clones within the population? If you can answer these questions, then one can begin to estimate how many integration sites would be required to claim that there are differences between proliferative clones and non-proliferative clones. Right now, there is a blind assertion that if it wasn't detected it was not a proliferative cell clone. At a very minimum, the limitations need to be spelled out and the name "non-proliferative" changed to a more nuanced title of undetected.

We thank the reviewer for their comments and address the reliability of our estimation of proliferative versus non-proliferative cells in the tissue organ based on our sampling of only a fraction of cells and counting the UMI per IS. To this end, we describe a mathematical analysis of how cells were sampled in an experiment to estimate the proportion of proliferating cells harboring provirus in different organs across various timepoints and animal subjects. In short, we found that even though some proliferative cells may be misclassified as non-proliferative (because of the small sample size), this typically only occurred for proliferative cells that were small clones, and thus comprise a small minority of all the proliferative cells in the organ. Consequently, the error introduced by the sampling methodology is small. Details of our findings including our quantification of the error are summarized below and described in full in **Supplementary Note 1 and Figures S6-9**.

Mathematical Analysis of the Sampling Procedure: Here we focused on understanding the sampling procedures using a dataset comprising of 48 different cases involving various animals, organs and three timepoints. Although we did not count every proliferating cell in each entire organ, we have measurements for the total number of cells (N_{tot}) and CD4 T cells (N_s) in the organ and the number of cells and CD4+ T cells harboring provirus (N_{tot} and N_s). It is assumed that the samples were well-mixed and represented the entire organ, meaning the proportion of proliferated infected CD4+ T cells in the sample is the representative of that organ. Thus, the total number of provirus-containing cells can be estimated as:

$$N_{tot} = N_s \frac{N_{tot}}{N_s}.$$

Numerical Approach to Estimate Proliferating Cells: To determine the error in estimating the percentage of proliferating cells in each experiment, we developed a numerical simulation method. Given the values of (N_{tot} and N_s), the fraction of proliferating cells in an organ, denoted as ν , is treated as a variable ranging between 0 and 1. The counts of proliferating and non-proliferating cells are defined as:

$$N_{prol} = \nu N_{tot}, \quad N_{non} = (1 - \nu) N_{tot}.$$

We assume the clone sizes of proliferative CD4+ T cells follow a power-law distribution¹:

$$g_n = Cn^{-\alpha}, \quad n = 2, 3, \dots, N_{tot}$$

Fig. S6: The distribution of UMI per IS.

a, The UMI per IS distribution in the two replicates from Figure S4c-3. Similar to our simulation and as described in the literature the proviral clone size (UMI/IS) follows a power-law distribution¹. The non-proliferative clone size was measured to be 93.6% in one sample, and 92.9% in the other sample.

b, The distribution of UMI per IS measurements in all the 48 experiments (combined). Here we assume the clone size of proliferative CD4 T cells obey a power-law distribution with the power $\alpha = 1$ ¹. Our simulation shows the distribution of the UMI per IS, which we measured and reflects the clonal composition of the organs.

where $\alpha = 1$ and C is a normalization constant. This distribution is consistent with observed clonal compositions measured by unique molecular identifiers (UMIs) per integration site, as shown in **Figure S6b**.

For a given fraction ν , we generate the clone sizes of proliferating cells (x_1, x_2, \dots, x_m) such that the sum of these clone sizes equals the total number of proliferating cells: $\sum_{i=1}^m x_i = N_{prol}$. Non-proliferating cells are treated as clones of size 1.

After simulating the sampling process N_s times, we calculate the counts of each clone's presence in the sample $y_1, y_2, \dots, y_{m+N_{non}}$. Clones represented only once ($y_i = 1$) are counted as by $N_{non}^{(s)}$, while clones represented more than once are counted as $N_{prol}^{(s)}$. The estimated fraction of proliferating cells is given by:

$$\nu^{(s)} = \frac{N_{prol}^{(s)}}{N_s}$$

Simulation Results: **Figure S7** displays typical simulation outcomes, showing predicted clone fractions versus true clone fractions for proliferative cells. Perfect predictions align along the diagonal line.

Figure S8 (shown in Supplementary Information) illustrates simulation results using nine values of the true fraction of proliferating cells ν ranging from 0.1 to 0.9. Each point represents 100 simulations, with the predicted values $\nu^{(s)}$ plotted against the true values. Results are shown for early infection, ART, and rebound infection timepoints.

Figure S9 summarizes these findings, indicating that the probability of accurately predicting the fraction of proliferating cells within 10% is 91%, 76%, and 99% for early infection, ART, and rebound infection, respectively. The average deviation from the true value is 0.8%, 2.3%, and 0.2% across these stages.

Thus, our mathematical analysis provides reliable estimates of proliferating cell fractions in most of the 48

experiments. Minor inaccuracies occur when the sample size is very small (about 1% of the total cells), but misclassified clones tend to be smaller and thus have a negligible impact on the overall predictions.

Fig. S9: A summary of the predicted vs “true” percentage of proliferating cells in the tissue. Average values of (N_{tot}, N_s) from samples during early infection, ART, and rebound infection are plotted.

The horizontal axis in each graph is v (true fraction) and the vertical axis is $v^{(s)}$ (predicted fraction). The symbols with vertical bars represent means and standard deviation from 100 simulations. The straight line is $v^{(s)} = v$.

integration sites, which were then subjected to deep sequencing to assess the reproducibility of our data. **Figure S4c** displays a weighted Venn diagram illustrating the overlap of barcodes between these two replicates. Among the shared proviral barcodes, the extent of proviral seeding per barcode (IS per BC) was also highly consistent and significantly correlated between the replicates, demonstrating that viral seeding patterns were remarkably similar in both samples (**Fig S4d**). Furthermore, the number of proviral barcodes between the replicates showed a strong and significant correlation, with a Pearson correlation coefficient of 0.97 ($p = 0.0048$) when analyzing the UMI per barcode or the copy number of proviral barcodes obtained from deep sequencing (**Fig. S4e**). **Figure S6a** shows the UMI per IS distribution in these two samples from Figure S4c-3. Similar to what we have seen, again, the proviral clone size (UMI/IS) follows a power-law

Fig. S7: Simulated sampling experiment, a typical run from spleen data during ART.

a, Predicted (y_i/N_s) clone fractions vs “true” (x_i/N_{tot}) clone fractions for proliferative cells. The diagonal line is $y_i/N_s = x_i/N_{tot}$, a perfect prediction. **b**, All the simulated clones sorted by size. The true clone fraction (blue) and the predicted clone fraction (orange) are plotted for each clone. The thin dashed line corresponds to the cases where only a single cell was present in the sample. Parameters are $N_{tot} = 2900$, $N_{prol} = N_{non} = N_{tot}/2$, $N_s = 339$, $\alpha = 1$.

In addition to the mathematical analysis and simulations presented above, we collected two independent DNA samples from the same organ of a rebounded animal. We used the BI-seq method described in this manuscript to construct libraries of proviral barcodes and

distribution. The non-proliferative clone size was measured to be 93.6% in one sample, and 92.9% in the other sample, which is very similar, and consistent with the simulation result.

We also assessed if viremia was more likely to be associated with proliferated cell clones as we reported in the manuscript. Indeed, in replicate 1, five out of six proliferated cell clones were associated with viremia and in replicate 2, all of the two proliferated cell clone were associated with viremia. This pattern was consistent with the manuscript, in which we found 27 out of 34

of the proliferated cell clones across all rebound samples were more likely to be associated with viremia (**Fig. 4a**).

After the statistical calculation, simulation, as well as performing experiments with real samples, we conclude that our methodology yields reliable results for the proviral clonal expansion estimation and the following viremia association study.

Other suggestions include:

The title of the manuscript is very limited to one part of the paper and doesn't even make sense (distinct chromatin patterns is not informative).

We appreciate the reviewer's suggestions and have changed the title to "Barcoded HIV-1 reveals viral persistence driven by clonal proliferation and distinct epigenetic patterns."

Throughout the paper, the use of "reseeding" to describe infection of cells is very confusing. Reseeding is poorly described word that denotes that the reservoir population is infected. But in most of the uses here, it is just describing infection. Line 64: "reseeding of proviruses by de novo infection of target cells and proliferation of cell clones" assumes that these infected cells will be the reservoir. And while the population in general doesn't change, most individual infected cells will die and not be the reservoir. I would highly recommend for readers that you use active replication pre-ART and ATI vs a block in replication during ART.

We have corrected the text, to describe de novo infection with establishment of proviral DNA as viral seeding, instead of the previous term of "viral reseeding" (lines 226-231). We have also

Fig. S4c-e: Barcode composition from two replicates sampling the same organ.

c, Weighted Venn diagram showing the overlapping of pooled barcodes in these two replicates. The number of proviral barcodes between the replicates showed a strong and significant correlation with a Pearson correlation coefficient of 0.97 ($p = 0.0048$).

d, Among the five overlapping proviral barcodes, the IS per BC for each barcode in replicates 1 and 2 were similar. P value was calculated by paired t-test.

e, Additionally, the IS per BC for each barcode in replicate 1 (x axis) and replicate 2 (y axis) were highly correlated. r represents Pearson's correlation coefficient with associated p value.

corrected the text to state the viral barcodes are seeded as proviral DNA (lines 518-520) as we agree with Reviewer 2 that is not synonymous to reservoir seeding.

Abstract: "despite perturbations such cellular clonal proliferation" should be "such as cellular"
Thank you, this has been corrected.

line 70: Not clear what in two individuals means. Two studies? Two donors of cells for humanized mice?

Thank you, this has been corrected to clarify two individuals living with HIV in line 70.

line 411: typo: "integration sties to the human genome"

Thank you, this has been corrected.

line 696: Should be "Blood and tissue processing"--not just tissues.

Thank you, this has been corrected.

line 162: acute infection (six weeks after HIV infection). Six weeks is well past acute infection. I would change that early or primary infection throughout the manuscript.

Thank you, we have changed acute infection to early infection throughout the manuscript.

line 165: "at intervals" isn't informative.

Thank you, this has been removed.

line 168: No data supporting the statement that engraftment was similar between groups.

The frequencies of human CD45+ cells among the groups of mice are now shown in Figure S2a. The p values were calculated using the one-way ANOVA test when 3 groups (early, ART, rebound) were compared right before HIV injection and then 6 weeks post HIV injection. The p values were calculated using the two-tailed Mann-Whitney test among the two remaining groups (ART and rebound) after 6 weeks of ART. These results show similar engraftment of human immune CD45+ cells between the groups.

line 169: list out organs at first use.

Thank you, this has been listed now as spleen, bone marrow, and human thymic implant.

line 171: "approximately 10⁴ copies per ml" I guess this is true after rebound, but not in primary infection. Should be specific here.

Thank you, we have revised the text in lines 178-180.

line 178: "In addition to tracking each viral lineage by barcode sequencing" prior to this statement, you haven't discussed barcode tracking. Not sure if this paragraph was in a different place originally, but currently out of place.

Thank you, that was indeed an error and we have revised the text in line 186.

lines 178-183: Very poor reasoning for changing methodologies. Limiting dilution works on samples with small total cells as well as with large numbers of cells. Not sure what you are trying to convey here. I think you can delete from 178-184.

We have revised lines 187-190 to clarify the reason for changing methodologies to, "Limiting dilution assays or single cell sequencing of rare sorted p24+ cells can be used to detect single molecule integration sites and their matching proviral sequences. However, translating methods based on limiting dilution assays to humanized mouse samples can be challenging due to the limited number of cells available. This often necessitates high-throughput, labor-intensive methods to avoid undersampling the reservoir."

lines 208-209: "Using BI-seq as a parallel platform, we examined a total of 293,557 viral DNA molecules from extracted genomic DNA of which 7.3% was integrated." I don't understand how so few genomes were integrated. This is way outside of normal for HIV in humans and SIV in NHPs. Is this common to this humanized mouse model? Please help readers understand this difference.

We apologize for the confusion. The frequency of integrated HIV has been shown to be a fraction of total DNA during both active viral replication and during ART in previous clinical studies^{10,11}. Prior studies have also shown using the Alu-PCR assay that integrated DNA levels ranged between ~11-66% of total DNA in clinical samples^{11,12}. Consistent with these studies, we found the frequency of integrated DNA out of total HIV DNA was approximately 3.2%, 20% and 9.0% during early infection, ART treatment, and rebound infection, respectively (**Rebuttal Fig. 3**).

Rebuttal Fig. 3: Pie charts of the percentage of integrated HIV vs other forms of HIV DNA (e.g. linear, circular, auto-integrated) during early infection, ART, and rebound infection.

line 218--"Persistent HIV infection can be driven by viral reseeding" this is not the conventional way we discuss viral replication. This and the reseeding score is unnecessarily complicated.

We have simplified the text in lines 226-227 to state, "Persistent HIV infection leads to de novo infection or clonal expansion of cells harboring proviral lineages."

line 282: "proliferated hundreds of times" is a huge underestimate. Be specific in the text: "detected hundreds of times" then you can say, which likely represents thousands of expanded cells. Again, if we knew more about the distribution of clonally expanded cells, we could be more specific on what it really means to detect a single clone "hundreds" of times.

Please see our response above in which we built statistical models, performed Monte-Carlo simulation, and also did experiments re-sampling of a biological duplicate of cells from the same organ pools to show that although sampling error does exist and misclassification of proliferated

and non-proliferative (because of the small sample size) can occur, this typically only occurs for small clones, and thus comprise a small minority of all the proliferative cells in the organ. Consequently, the error introduced by the sampling methodology is small. Details of our findings including the quantification of the error are summarized above.

line 291: "The decrease in the cell proliferation index during rebound infection". These data and the ensuing rationalization wasn't clear to me. Viral load was very high so lower proliferation index would suggest that replication in non-proliferating cells was the reason for high viral load which is not seen in primary infection. So is there a reason for this or not.

We found smaller proliferated cell clone sizes were detected during rebound infection (**Fig. 3e**). As reviewer 2 suggests this could indicate that proliferated cell clones were not contributing to rebound viremia. However, we found multiple results indicating that proliferated cell clones were associated with rebound viremia. For example, we matched the proviral barcodes to the viral RNA barcodes in rebound viremia, and found that proliferated cell clones were 19.7 times more likely to be associated with rebound viremia compared to non-proliferated cell clones (**Fig. 4b**). In a similar trend, proliferated cell clones were 4.2 times more likely to be associated with viremia during early infection (**Fig. 4c**). Consistent with these results, we found proliferated cell clones had proviruses were significantly more likely to have chromatin features suggesting proviral activation compared to non-proliferated cell clones (**Fig. 6**). Furthermore, proliferated cell clones associated with viremia had proviruses with more activating chromatin marks compared to proliferated cell clones not associated with viremia (**Fig. 7**). Interestingly, we found proliferated cell clones associated with viremia were smaller in size compared to those that were not associated with viremia (**Fig. 4d**). Indeed, we found a negative correlation between the cell clone size (UMI per IS) and the extent of viral spread (UMI per BC) (**Fig. 4e**). Collectively these results indicate proliferated cell clones were highly associated with rebound viremia, but not large cell clone sizes. Future studies can investigate why smaller proliferated cell clones rather than larger ones are associated with rebound viremia. We suspect that cellular clones that contribute to rebound viremia may be smaller because the rate of cellular proliferation does not completely outcompete the elimination of transcriptionally active cells producing virus.

lines 411 and 412: proliferated versus non-proliferated clones were 1.1-fold more likely to be in genes. Figures 6 and 7 show significance with very small magnitude changes. I am not a statistician, but it does not pass the common sense test.

Reviewer 2 highlights that the odds ratio (OR) of finding integration sites in genic regions among proliferated versus non-proliferated cell clones was 1.1 (**Fig. 6A, right**). We agree that an OR of 1.1 could be interpreted as small magnitude change in certain circumstances, but 10% increases could also be meaningful in other biological situations. Additionally, the OR of 1.1 was associated with a p value that was significant as calculated by Fisher's exact test. The reason we are having a small yet significant OR could be that the human genome location for HIV integration is highly heterogeneous. In addition, the fate of a T cell clone can be influenced by a variety of other factors, such as the specific types of T cells or what developmental program and cell cycle stages the T cell is at during infection. All these factors contribute to a lower OR of the association of an integration into a genic location and the proviral clone being proliferative. In addition, we also listed the OR of finding proviruses in multiple sites associated with transcriptional activation (e.g.

genic regions, genome regulation elements such as promoters, enhancers, CTCF or TF binding sites) and their associated histone marks. Thus, Figures 6 and 7 analyze multiple genic and epigenetic marks with significant odds ratios ranging from 1.1 to 4.12 to show a significant overall trend in which proviruses in proliferated cell clones were significantly correlated with chromatin features associated with proviral activation.

Reviewer #3 (Remarks to the Author):

Main comments

This is an interesting paper which describes the development of a new and highly precise strategy to study the HIV1 reservoir in a humanized mouse model. The Authors have confirmed previous findings from other studies in which the clonal expansion of HIV infected clones and, importantly, show a complex interplay between integration in chromatin active or active genomic regions leading to different expression levels of viral genes, counter-selection caused by toxicity of viral infection or immunological counter selection but in different contexts selection of proliferating infectious clones during the acute infection. Overall, this is a well written manuscript with interesting findings on HIV persistence in vivo and the factors impacting on reseeding and genomic integration during acute infection, under ART and after ART interruption. Importantly this experimental methodology has the potential to expand our knowledge on HIV1 biology, study the effects of alternative ART or additional/alternative pharmacological interventions aimed at the permanent HIV eradication.

Few minor suggestions are:

1) The strategy to analyze the HIV repertoire by the study of barcodes and integration sites from DNA and barcodes on viral RNA in vivo is quite complex for the non-aficionados. The experimental strategy should be explained in a graphical way expanding Figure 1 A. I believe that the inclusion of a graphical representation of the experimental strategy would be helpful for the readers.

We thank Reviewer 3 for their insightful suggestion and have now included a graphical abstract depicting the overall experimental workflow to detect either viral RNA barcodes and viral DNA barcode PCR linked to integration sites from an organ sample. **Figure S3b** shows a graphical abstract for the workflow of how the viral DNA barcode is PCR-linked to the integration site.

2) During Illumina sequencing it is possible to find some barcoded oligonucleotides swapped across samples. This phenomenon, which occurs during bridge PCR, does not pose great issues for the sample recognition as wrong sample barcode combinations are easily eliminated bioinformatically. However, this phenomenon could artificially inflate the clonal abundances of specific integrations because different UMIs could be included by the swapping phenomenon. In this case the analysis of the coherence of samples barcodes and integration site will not help. Did the Authors measure the extent of this phenomenon in their datasets? And how to deal with such issue (if present)?

We thank the reviewer for bringing up a very important question regarding index hopping during bridge PCR in deep sequencing. The frequency of index hopping can occur from <0.5% to up to 10%, depending on the platform and methods of estimation. Indeed, we observed index hopping in our previous deep sequencing libraries, where UMI overlap was significantly higher in libraries that share one index barcode. We changed our index barcode setup to ensure each sublibrary contains unique barcodes in both the i5 and the i7 end. In addition, the i5 index and i7 index barcodes are at least three nucleotide differences among each other. In this manner, a majority of molecules that were index hopped during bridge PCR were removed during the demultiplexing step and excluded from analysis. Another feature of the BI-seq platform is that each UMI (referred as primerID in the following text) represents a single molecule in the beginning pool, such that even if some primerID accidentally spilled over to adjacent samples and dominant during the following amplification steps, it can be removed from all samples without interfering the total integration site and barcode analysis. In fact, by removing the most dominant 454 primerIDs from all the >2e4 primerIDs, that is, removing <3% molecules from the deep sequencing, our UMI cross contamination is estimated to be 1.61% on average (CI95 0.0150 - 0.0172).

In terms of whether index hopping (UMI included into another libraries) will cause over-estimation of clonal abundances of specific integrations, here's our quality control step. We quantify the size of each integration site clone by counting the number of unique primerIDs associated with it. Since our primerID is of 20 nucleotide long, which can have a diversity of 420, that is over 1e12 complex, the chances of two molecules sharing the same primerID is infinitesimal. Therefore, as long as we can confidently assign the linkage of an integration site to a primerID, the estimation of the clone size of a provirus will not be inflated even in the presence of index hopping. We optimized our IB sequencing protocol with a quality control step to look at the primerID-integration site association confidence.

The integration site confidence score is expressed as the count of the most dominant site associated with that primerID over the total counts of integration sites associated with that primerID. Ideally there should only be one integration site associated with one primerID, also referred henceforth as a confidence score equal to 1. However, at times we can still see some minor species formed. If the confidence score is over 0.6, meaning more than 60% of the time, a primerID can be assigned to one integration site, we consider that association confident and remove the non-dominant associations. Shown in **Rebuttal Fig. 4** is the overall plot for all the primerIDs in our dataset, x axis represents each primerID's frequency and y axis represents the confidence score.

Rebuttal Fig. 4: Scatter plot of frequency and integration site linkage confidence for each primerID in the whole dataset.

As we can see from **Rebuttal Fig. 5**, primerID with low confidence scores are those that have low sequencing depth. Overall, with our IB sequencing protocol, we have >95.68% of the primerID in our dataset (read count > 3) that are confidently linked to an integration site with a confidence score >0.8. Therefore, we concluded that such swapping does exist in our dataset but takes up a very small fraction for only low reading depth primerIDs, and we can confidently correct most of them so as not to interfere with our downstream analysis.

3) In the field on integration site analysis, shear site statistics and statistical modelling algorithms such as “soniclength” (Berry, C. C. et al. Estimating Abundances of Retroviral Insertion Sites from DNA Fragment Length Data. Bioinformatics 2012) have been used with success. It would be interesting if the two methods could be compared.

We thank the reviewer for bringing up an interesting comparison of another method that has been used widely to quantify integration sites in a population of cells, that is, fragmenting the genome using a randomized method (such as sonication) and estimating the abundance of each integration site by counting the unique length associated with it. The non-linearity of the relation between number of unique lengths and relative abundance can be addressed by statistical modeling, as the reference paper shown by the reviewer. In comparison, we choose to fragment the genome by using a type II restriction Enzyme (HinP1i) that recognizes a four nucleotide sequence. Then,

Rebuttal Fig. 5: Fraction of primerID with confidence score over 0.6 at different read count filter cutoff.

Rebuttal Fig. 6: Distribution of human genome fragment sizes virtually digested by indicated enzymes.

we ligated an adaptor with a UMI and quantified the abundance of each integration site by counting the UMI. The method we chose has several benefits: 1) Using a restriction enzyme digestion method instead of sonication or fragmentase is easier to handle over samples with small volume (in the case of sonication larger volumes are needed) and consistency over different samples; 2) By choosing an enzyme that has no cut site within the amplified region containing viral barcode, we will maximize the probability of retrieving proviral barcodes in our BI-seq platform; 3) The proviral copy number across the whole infection course falls below $1e4$ copies, which can be accurately measured by deep sequencing, and the counting of UMI can be directly converted to the proviral copy number as shown in our **Extended Data Fig. 3c**. What we found is that a significant portion of provirus were in cell lineages that do not undergo clonal expansion (only a single copy of provirus exists in the pool). Using restriction enzymes to achieve complete digestion followed by several rounds of on target amplification will give us a higher chance to preserve these species in the final sequencing pool. Counting the unique length generated by random fragmentation, on the other hand, even with the help of bioinformatics to accurately estimate the relative abundance of integration sites by conversion to direct copy number of the proviral molecule is still not straightforward.

However, we also acknowledge that there are some disadvantages associated with our method of using restriction enzymes. The most obvious one would be that the integration sites may not contain any cutting recognition site or too many cutting recognition sites to be detected accurately by our method. Thus, the estimation of the diversity of integration sites could be influenced by the choice of enzymes. To this end, we performed pilot experiments to compare the sensitivity of different restriction enzyme choices.

We tested four types of digestion: BamHI+EcoRI double digestion, HinP1i, Hpy166II, and HpyCH4III. All of the four types of digestions generated an average fragment between $1e2$ to $1e4$ bp in length. The distribution of human genome cut by these four types of digestions are shown in **Rebuttal Fig. 6**.

Rebuttal Fig. 7: Sensitivity test of BI-seq libraries generated with different restriction enzyme during the fragmentation step. Known amount of provirus were subject to the indicated restriction enzymes and the following BI-seq library preparation and read count were retrieved by deep sequencing.

Next, we quantified the sensitivity of these four scenarios by inputting different amounts of proviruses. Out of these four scenarios, HinP1i gave the highest sensitivity, that is, it detected single digits of provirus in the pool. Therefore, we used HinP1i in our BI-seq platform.

In conclusion, our method of using restriction enzymes to fragment the genome and primerID to quantify each provirus copy number is highly sensitive. In addition, our choice of the HinP1i enzyme was highly tailored to suit our barcode NFNSX virus. In other situations (such as other types of retrovirus) where integration sites need to be quantified, the choice of restriction site enzyme and protocol will require empirical testing.

4) The finding that some integrations in expanded clones targeted oncogenes is interesting and it should be expanded in the discussion. Indeed, although these integrations were rare, they are probably also relevant especially considering the finding that the expanded clones have an important role in the maintenance of the complexity of the clonal reservoir. On this regard it is interesting to note that the Authors did not find enrichment of integration sites targeting BACH2 nor STAT5B. Since these integrations appeared to be enriched in long lived and fully immunosuppressive T-regulatory cells it may be difficult to identify them in this mouse model. Could the Authors elaborate on this aspect? Do these humanized mice produce a detectable number of T-regulatory cells?

We thank Reviewer 3 for their insightful comments. We have further clarified our results and indicate that proviruses were detected in the BACH2 and the STAT genes, but not necessarily significantly favored among the expanded cell clones compared to non-proliferated cell clones (**Supplementary Table S4**). Although we did not find BACH2 and STAT genes were specifically enriched among our proliferated cell clones, we did find five other genes associated with tumor progression, cell proliferation and anti-apoptosis were significantly enriched (e.g. WWC1, TNFRS10D, FOXP4, TNFRS10B, NBPF12) (**Fig. 5c**). Interestingly, we also found TNFRS10D and FOXP4 were not only enriched but significantly associated with larger cell clone size (UMI per IS) (**Fig. 5e**). It is well-known that integration is favored among actively transcribing genes. Thus, enrichment of integration into TNFRS10 (also known as TRAIL-R2 or Death Receptor 5) and FOXP4 among the large, proliferated cell clones could suggest a role for activated CD4+ T cells and/or T regulatory cells (Tregs) in proviral persistence (as the reviewer astutely suggests). Tregs generally exhibit resistance to apoptosis, allowing them to survive and proliferate compared to other T cell subsets. Additionally, the immunosuppressive function of Tregs could protect them for immune mediated clearance. Interestingly, humanized mice have similar levels of Tregs in their human thymic implant compared to normal human fetus and detectable levels in the blood although they tend to have a more naïve phenotype (CD4⁺CD25⁺FOXP3⁺) compared to normal human PBMCs¹³. Thus, we agree with Reviewer 3 that future studies should investigate whether Tregs may be involved in large, expanded cell clones harboring provirus in our model. However, because overall we found only a small fraction of proliferated cell clones had integration sites enriched in cell survival genes (**Fig. 5d**), we tend to agree with the study from Coffin J, et al. that positive selection for a few oncogenes likely only modestly reshapes proviral distribution in vivo¹⁴.

Rebuttal References

- 1 Gaimann, M. U., Nguyen, M., Desponds, J. & Mayer, A. Early life imprints the hierarchy of T cell clone sizes. *Elife* **9** (2020). <https://doi.org:10.7554/eLife.61639>

- 2 Marsden, M. D. *et al.* In vivo activation of latent HIV with a synthetic bryostatin analog effects both latent cell "kick" and "kill" in strategy for virus eradication. *PLoS Pathog* **13**, e1006575 (2017). <https://doi.org:10.1371/journal.ppat.1006575>
- 3 Denton, P. W. *et al.* Generation of HIV latency in humanized BLT mice. *J Virol* **86**, 630-634 (2012). <https://doi.org:10.1128/JVI.06120-11>
- 4 Marsden, M. D. *et al.* HIV latency in the humanized BLT mouse. *J Virol* **86**, 339-347 (2012). <https://doi.org:10.1128/JVI.06366-11>
- 5 Lavender, K. J. *et al.* An advanced BLT-humanized mouse model for extended HIV-1 cure studies. *AIDS* **32**, 1-10 (2018). <https://doi.org:10.1097/QAD.0000000000001674>
- 6 Kim, J. T. *et al.* Latency reversal plus natural killer cells diminish HIV reservoir in vivo. *Nat Commun* **13**, 121 (2022). <https://doi.org:10.1038/s41467-021-27647-0>
- 7 Rajashekar, J. K. *et al.* Modulating HIV-1 envelope glycoprotein conformation to decrease the HIV-1 reservoir. *Cell Host Microbe* **29**, 904-916 e906 (2021). <https://doi.org:10.1016/j.chom.2021.04.014>
- 8 Dudek, T. E. *et al.* Rapid evolution of HIV-1 to functional CD8(+) T cell responses in humanized BLT mice. *Sci Transl Med* **4**, 143ra198 (2012). <https://doi.org:10.1126/scitranslmed.3003984>
- 9 Claiborne, D. T. *et al.* Immunization of BLT Humanized Mice Redirects T Cell Responses to Gag and Reduces Acute HIV-1 Viremia. *J Virol* **93** (2019). <https://doi.org:10.1128/JVI.00814-19>
- 10 Furtado, M. R. *et al.* Persistence of HIV-1 transcription in peripheral-blood mononuclear cells in patients receiving potent antiretroviral therapy. *N Engl J Med* **340**, 1614-1622 (1999). <https://doi.org:10.1056/NEJM199905273402102>
- 11 Agosto, L. M. *et al.* Patients on HAART often have an excess of unintegrated HIV DNA: implications for monitoring reservoirs. *Virology* **409**, 46-53 (2011). <https://doi.org:10.1016/j.virol.2010.08.024>
- 12 Yu, J. J. *et al.* A more precise HIV integration assay designed to detect small differences finds lower levels of integrated DNA in HAART treated patients. *Virology* **379**, 78-86 (2008). <https://doi.org:10.1016/j.virol.2008.05.030>
- 13 Onoe, T. *et al.* Human natural regulatory T cell development, suppressive function, and postthymic maturation in a humanized mouse model. *J Immunol* **187**, 3895-3903 (2011). <https://doi.org:10.4049/jimmunol.1100394>
- 14 Coffin, J. M. *et al.* Integration in oncogenes plays only a minor role in determining the in vivo distribution of HIV integration sites before or during suppressive antiretroviral therapy. *PLoS Pathog* **17**, e1009141 (2021). <https://doi.org:10.1371/journal.ppat.1009141>

Dear Editor,

We appreciate the opportunity to respond to the reviewers' comments for our manuscript NCOMMS-24-32622A to Nature Communications. We appreciate the insightful critique from the reviewers and the editorial staff during the peer review process to strengthen the manuscript. Our comments to the reviewers are in blue below.

Remaining reviewer comments:

Reviewer #1:

The authors have addressed my previous concerns

Thank you Reviewer 1 for your time and consideration in strengthening our manuscript during the peer review process.

Reviewer #2:

Congratulations on a comprehensive and highly useful update to your manuscript. I was happy to see the changes that were made and I have no further questions or concerns.

Thank you Reviewer 2 for your time and consideration in strengthening our manuscript during the peer review process. We are also happy with the changes to the manuscript yielded during the peer review process.

Reviewer #3:

Author's response has addressed my points

Thank you Reviewer 3 for your time and consideration in strengthening our manuscript during the peer review process.